



# Isotopic diffusion in ice enhanced by vein-water flow

Felix S. L. Ng[1]

[1]Department of Geography, University of Sheffield, Sheffield, UK

*Correspondence to*: Felix Ng (f.ng@sheffield.ac.uk)

**Abstract.** Diffusive smoothing of signals on the water stable isotopes ($^{18}$O and D) in ice sheets fundamentally limits the climatic information retrievable from these ice-core proxies. Past theories explained how, in polycrystalline ice below the firn, fast diffusion in the network of intergranular water veins "short-circuits" the slow diffusion within crystal grains to cause "excess diffusion", enhancing the rate of signal smoothing above that implied by self-diffusion in ice monocrystals. But the controls of excess diffusion are far from fully understood. Here, modelling shows that water flow in the veins amplifies excess diffusion, by altering the three-dimensional field of isotope concentration and isotope transfer between veins and grains. The rate of signal smoothing depends not only on temperature, vein and grain sizes, and signal wavelength, but also on vein-water flow velocity, which can increase the rate by 1 to 2 orders of magnitude. This modulation can significantly impact signal smoothing at ice-core sites in Greenland and Antarctica, as shown by simulations for the GRIP and EPICA Dome C sites, which reveal sensitive modulation of their diffusion-length profiles when vein-flow velocities reach $\sim 10^1$–$10^2$ m yr$^{-1}$.

Velocities of this magnitude also produce the levels of excess diffusion inferred by previous studies for the Holocene ice at GRIP and ice of Marine Isotope Stage 19 at EPICA Dome C. Thus, vein-flow mediated excess diffusion may help explain the mismatch between modelled and spectrally-derived diffusion lengths in other ice cores. We also show that excess diffusion biases the spectral estimation of diffusion lengths from isotopic signals (by making them dependent on signal wavelength) and the reconstruction of surface temperature from diffusion-length profiles (by increasing the ice contribution to diffusion length below the firn). Our findings caution against using the monocrystal isotopic diffusivity to represent the bulk-ice diffusivity. The need to predict the pattern of excess diffusion in ice cores calls for systematic study of isotope records for its occurrence and improved understanding of vein-scale hydrology in ice sheets.





## 1 Introduction

The water stable isotope records ($\delta^{18}$O, $\delta$D) from ice cores are key proxies for reconstructing palaeoclimatic temperature.

Isotope diffusion, which occurs rapidly in firn (mainly by vapour diffusion in the pores) and slowly in ice below the firn transition, causes progressive smoothing that reduces the high-frequency content of these records, strongly attenuating the amplitude of short signals, and limiting the depth to which annual and seasonal information survives (Johnsen, 1977; Whillans and Grootes, 1985; Cuffey and Steig, 1998, Johnsen et al., 2000). In the ice cores from central Greenland, annual signals often persist for $\sim 10^4$ years, to the early Holocene or the late-glacial part of the record (Johnsen, 1977; Johnsen et al., 1997; Johnsen et al., 2000), whereas in the ice cores from the East Antarctic plateau, they rarely penetrate through the firn into the ice, owing to low accumulation rates (causing short $\delta$-cycles, which decay quickly) and substantial noise and intermittency on the signals during deposition (Laepple et al., 2018; Casado et al., 2020).

Post-depositional diffusive smoothing limits the time resolution of real climate signals extractable from different depths of an ice-core isotope record. The smoothing rate needs to be known in several analyses: (i) studies that use "back-diffusion" or deconvolution (Johnsen 1977; Johnsen et al., 2000) to restore the original annual $\delta$-cycles at the surface, for inferring detailed climatic variations (e.g., Küttel et al., 2012; Zheng et al., 2018) or aiding the identification of annual layers in ice-core dating (e.g., Hammer et al., 1978; Vinther et al., 2006); (ii) studies that use the diffusion length $\sigma$ estimated from the frequency spectrum of isotopic signals at different depths (including where annual cycles are no longer visible) to determine past changes in $\sigma$ at the firn transition, and hence reconstruct the surface temperature history by using the temperature dependence of firn isotope diffusion (e.g., Gkinis et al., 2014; Holme et al., 2018; Kahle et al., 2021); and (iii) studies that model the down-core profile of the diffusion length to assess the climatic variability on different time scales on a record (e.g., Jones et al., 2017; Gkinis et al., 2021; Grisart et al., 2022). The diffusion-length theory of Johnsen (1977), which tracks how $\sigma$ evolves as a result of isotopic diffusion and vertical compression in the ice column, forms the basis of all these studies.

Whereas the smoothing process in firn has been studied sufficiently to yield models for the temperature reconstructions (in (ii) above) and able to reproduce the observed signal decay in firn (e.g., Cuffey and Steig, 1998; Johnsen et al., 2000; Gkinis et al., 2021), the smoothing process in ice remains poorly understood. We extend its theoretical description in this paper. Diffusive smoothing in ice can strongly impact deep isotopic signals given their long residence time; also, the diffusion rate increases in the warmer ice towards the ice-sheet base. Thus, a key concern motivating our work is that an inaccurate model for the signal smoothing rate in ice can bias the aforementioned studies – notably studies of types (ii) and (iii) above, when applied to records far below the firn transition. This problem may extend to reconstructions that use the differential diffusion length between oxygen and deuterium as a temperature proxy (Simonsen et al., 2011; Holme et al., 2018).



From the decay of annual $\delta^{18}$O cycles along the Holocene part of the GRIP ice core, Johnsen et al. (1997) inferred an
isotopic diffusion rate about 10 times faster[1] than the self-diffusion rate in ice monocrystals (Ramseier, 1967), measured at the
temperature of the GRIP ice under analysis. They suggested the grain interfaces in polycrystalline ice as causing "excess
diffusion" – an idea which prompted three mathematical models seeking to explain the phenomenon. Nye (1998) modelled the
effect of water veins located at the triple junctions of grain boundaries (Nye, 1989; Mader, 1992a, b) and showed how rapid
(liquid-phase) diffusion in the vein network "short-circuits" slow diffusion in the ice grains to enhance the signal-decay rate
above that due to solid diffusion. For signals at the decimetre scale, and ice with a mean grain size of several millimetres, his
model predicts an enhancement that matches the GRIP observations. Johnsen et al. (2000) considered more generally
interstitial water at grain boundaries as well as in the veins and calculated how much these pathways raise the effective isotope
diffusivity of the bulk ice. Their model couples the isotope concentrations in the solid and liquid in a less sophisticated way
than Nye's treatment, but accounts for the tortuosity of the veins, and they highlight the possibility for the acidity of the ice to
affect the amount of interstitial water. Lastly, Rempel and Wettlaufer (2003), building upon Nye's (1998) continuum
description of the grain–vein system, showed that the perfect short-circuiting assumed in Nye's model overestimates the level
of excess diffusion: the enhancement is less than the value predicted by Nye as the liquid diffusivity is high but finite. Rempel
and Wettlaufer (2003) clarified the two previous models as end-member approximations of the system; and like Nye's result,
their solution gives the enhancement as a function of signal wavelength. All three models – of Nye, Johnsen et al. and Rempel
and Wettlaufer – predict a higher enhancement for thicker veins, because wider liquid pathways promote short-circuiting.

Despite these models' implication that recrystallisation and impurity processes in polycrystalline ice can alter the amount
of excess diffusion to shape isotope records in complex ways, no diffusion-length based models or temperature reconstructions
have yet incorporated their results into calculations, which typically assume the monocrystal diffusivity of Ramseier (1967)
below the firn transition. Nor has there been progress in unravelling the controls and mechanisms of excess diffusion – by
theory or experiment – for two decades. Yet, the potential occurrence of excess diffusion continues to concern ice-core studies.
When analysing the WAIS Divide ice core, Jones et al. (2017) invoked excess diffusion as one of several explanations why
diffusion lengths derived from $\delta$D signals at $\approx$ 15–18 ka BP exceeded modelled diffusion lengths based on Ramseier's

diffusivity by up to 1.6 times. For the high-resolution $\delta$D record of the EPICA Dome C core, Pol et al. (2010) regarded vein-

driven enhancement of isotopic diffusion to be the cause of strong smoothing and near absence of sub-millennial scale signals

across Marine Isotope Stage 19 (MIS 19) – the oldest interglacial identified in that core, at ~780 ka BP – where spectrally-

derived diffusion lengths (~40 to 60 cm) are several times higher than predicted by monocrystal diffusivity. Excess diffusion

is expected to impact the preservation of deep isotopic signals in the ice cores to be retrieved at Little Dome C, Antarctica, by

---

[1] Recounting the same analysis, Johnsen et al. (2000) later reported 30 times.



the European Beyond EPICA project and the Australian Million Year Ice Core project, which aim to obtain records reaching back $\sim$ 1–1.5 Ma and covering the Mid-Pleistocene Transition.

Herein, we revisit Nye's (1998) and Rempel and Wettlaufer's (2003) formulation for vein-mediated excess diffusion, asking "what if the vein water isn't stagnant, but percolates?" In our modelling in Sects. 2 and 3, we show that vein-water flow distorts the isotopic concentrations in ice grains and modifies their isotope exchange with the veins, so that excess diffusion acting on isotopic signals is always increased from the no-flow case. The mechanism causes signals to move relative to the ice also, although the age offset of displaced signals is much less than their absolute age. Depending on the water flow velocity,

the decay-rate enhancement for decimetre-scale signals can vary by a factor of several to a few hundred – between Rempel and Wettlaufer's and Nye's predictions. This modulation highlights the vein hydrology of ice sheets as a major knowledge gap. In Sect. 4, we explore its impact on signal smoothing at ice-core sites by embedding it in diffusion-length simulations for the GRIP and EPICA ice cores. We show that excess diffusion can undermine diffusion-based temperature reconstructions and the spectral derivation of diffusion lengths from isotope records. We conclude with broader perspectives in Sect. 5.

## 2 Mathematical model

As in Nye's (1998) and Rempel and Wettlaufer's (2003) studies, our modelling in this section focusses on the interactions between crystals and veins at the grain scale, ignoring the effect of ice deformation on isotopic signals, and ignoring diffusion along grain boundaries. Vertical compression will be accounted for in Sect. 4. Background about the diffusion-length theory will be given there.

First we extend their equations to incorporate vein-water flow. We adopt their idealised geometrical set up (Fig. 1), which represents ice crystal grains as a vertical annular cylinder, with outer radius $b$, and the water vein as a hole at its centre, with the vein wall located at the inner radius, $r = a$ ($\sim 10^{-6}$ m); $r$ is the radial coordinate. We distinguish the vein radius $a$ from the radius of curvature of real (convex) vein walls (Nye, 1989; Mader, 1992a; Ng, 2021). In plan view, each cylinder is meant to approximate a unit cell of polycrystalline ice around a vein, so $b$ is taken as the mean grain radius ($\sim 10^{-3}$ m). As the original

studies assumed, the vein water is kept liquid by a high concentration of dissolved ionic impurities, which lowers the eutectic temperature; and horizontal (/near-horizontal) veins are disregarded as they cause no (/negligible) short-circuiting of depth-varying isotope signals. With the $z$ coordinate axis pointing down, and $t$ denoting time, the concentrations of a trace isotope ($^{18}$O or D) in the ice grains and in the vein water – $N_s(r, z, t)$ and $N_v(z, t)$, respectively – satisfy the conservation equations

$$\frac{\partial N_s}{\partial t} = D_s \left( \frac{1}{r} \frac{\partial}{\partial r} \left( r \frac{\partial N_s}{\partial r} \right) + \frac{\partial^2 N_s}{\partial z^2} \right) , \tag{1}$$

$$\frac{\partial N_v}{\partial t} = D_v \frac{\partial^2 N_v}{\partial z^2} + \frac{2 D_s}{a} \left. \frac{\partial N_s}{\partial r} \right|_{r=a} - w \frac{\partial N_v}{\partial z} , \tag{2}$$





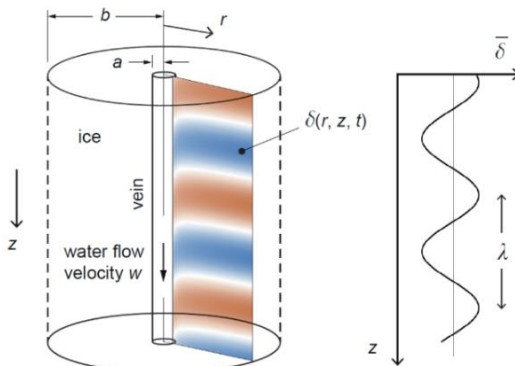

Figure 1. Model geometry and symbols. Left: ice annular cylinder surrounding a vein. Colour image shows a radial cross-section of the isotopic deviation $\delta$ in the ice, exemplifying the signals studied in Figs. 3 to 5. The pattern is distorted by the boundary condition at the vein wall due to isotope advection by vein-water flow. Right: depth profile of the mean isotopic signal.


where $D_s$ and $D_v$ are molecular diffusivities in the solid (single crystal) and water. Following the original studies, $N_v$ is assumed independent of $r$, and we specify $\partial N_s/\partial r = 0$ at $r = b$ as boundary condition. In Eq. (2), which couples $N_s$ and $N_v$, the $D_s$ term represents isotope transfer between ice and vein. The final term – our addition to the model – describes advection of $N_v$ by

vein water flowing at velocity $w$ (positive downward).

Equilibrium fractionation at the ice-water interface implies $\alpha N_v/N_{v0} = N_s|_{r=a}/N_{s0}$, where $\alpha$ ($\approx 1$) is the fractionation coefficient, and $N_{v0}$ and $N_{s0}$ (assumed constant) are the number densities of the major isotope ($^{16}$O or H) in water and ice. Following the procedure of Rempel and Wettlaufer (2003), we rewrite $N_v$ in Eq. (2) in terms of $N_s|_{r=a}$, assuming $N_{v0} \approx N_{s0}$, and express $N_s$ as the isotopic deviation $\delta = \delta(r, z, t) = N_s/N_{s0} - 1$, thus deriving

$$\frac{\partial \delta}{\partial t} = D_s \left( \frac{1}{r} \frac{\partial}{\partial r} \left( r \frac{\partial \delta}{\partial r} \right) + \frac{\partial^2 \delta}{\partial z^2} \right) \; , \tag{3}$$

with the boundary conditions

$$\left. \frac{\partial \delta}{\partial r} \right|_{r=b} = 0 \tag{4}$$

and

$$\left. \frac{\partial^2 \delta}{\partial r^2} \right|_{r=a} + \frac{1-2\alpha}{a} \left. \frac{\partial \delta}{\partial r} \right|_{r=a} - \beta \left. \frac{\partial^2 \delta}{\partial z^2} \right|_{r=a} + \frac{w}{D_s} \left. \frac{\partial \delta}{\partial z} \right|_{r=a} = 0 \; . \tag{5}$$

The second boundary condition has been derived by eliminating the time derivatives between (1) and (2). The dimensionless parameter





$$\beta = \frac{D_v}{D_s} - 1 \quad (> 0) \tag{6}$$

quantifies the diffusivity contrast of water to ice. The diffusivities $D_s$ and $D_v$ vary strongly with temperature $T$, and typically $\beta \sim 10^6$ (Fig. 2). As detailed in Appendix A, we use Ramseier's (1967) formula for $D_s(T)$, and for $D_v(T)$ we use an extension

of Gillen et al.'s (1972) formula that is valid down to –60°C.

Equations (3) to (5) form a partial differential equation model for $\delta(r, z, t)$. Rempel and Wettlaufer (2003) assumed $1-2\alpha \approx -1$ in Eq. (5), so their model is independent of the fractionation coefficient $\alpha$ and applies equally to $\delta^{18}$O and $\delta$D. This is a good approximation because $\alpha(^{18}\text{O}/^{16}\text{O}) \approx 1.0029$ and $\alpha(\text{D/H}) \approx 1.021$ (Lehman and Siegenthaler, 1991; O'Neill, 1968; Árnason, 1969). We make the same approximation in most of Sects. 3 and 4, but not in the present derivation, as we need a

general model that observes the precise value of $\alpha$, for an analysis about dual-isotope thermometry at the end of Sect. 4.

Equations (3) to (5) encapsulate the short-circuiting effect and differ from Rempel and Wettlaufer's (2003) model by the $w$-term only. When studying the system without water flow ($w = 0$), these authors explained that Nye's (1998) model corresponds to the limit $\beta \to \infty$, as it supposes liquid diffusion so fast that $\delta$ along the vein is constant (i.e., perfect short-circuiting); on the other hand, the model of Johnsen et al. (2000) effectively assumes instantaneous radial diffusion in the grains, so that

longitudinal diffusion along the vein and in the ice governs the smoothing of signals. Thus the Johnsen et al. model predicts an excess diffusion equal to Rempel and Wettlaufer's prediction for slow-varying (long) signals, but is not strictly an approximation of our model at a limit of a defined parameter here. We shall not compare its predictions against our results.

When vein-water flow occurs ($w \neq 0$), we expect advection of $\delta$ at the vein boundary to perturb $\delta$ in the ice (Fig. 1), causing the isotopic signals there to move also. To study how signals behave, we seek a separable solution of the form

$$\delta(r, z, t) = \delta_0 + \delta_1 F(r) \exp(-D_s \zeta t + i k_z z), \tag{7}$$

where $\zeta = \zeta_R + i\zeta_I$ is a decay-rate parameter for sinusoidal signals with the wavenumber $k_z$ (or wavelength $\lambda = 2\pi/k_z$), and $\delta_0$ and $\delta_1$ are arbitrary constants representing the background level and amplitude of signals. The amplitude of signals decays at the rate $D_s \zeta_R$, whereas their 'baseline' decay rate in ice without veins (due to solid diffusion alone) would be $D_s k_z^2$. Following Nye (1998) and Rempel and Wettlaufer (2003), we let

$$\zeta_R = k_z^2 + k_r^2 = f k_z^2, \tag{8}$$

in which the "enhancement factor"


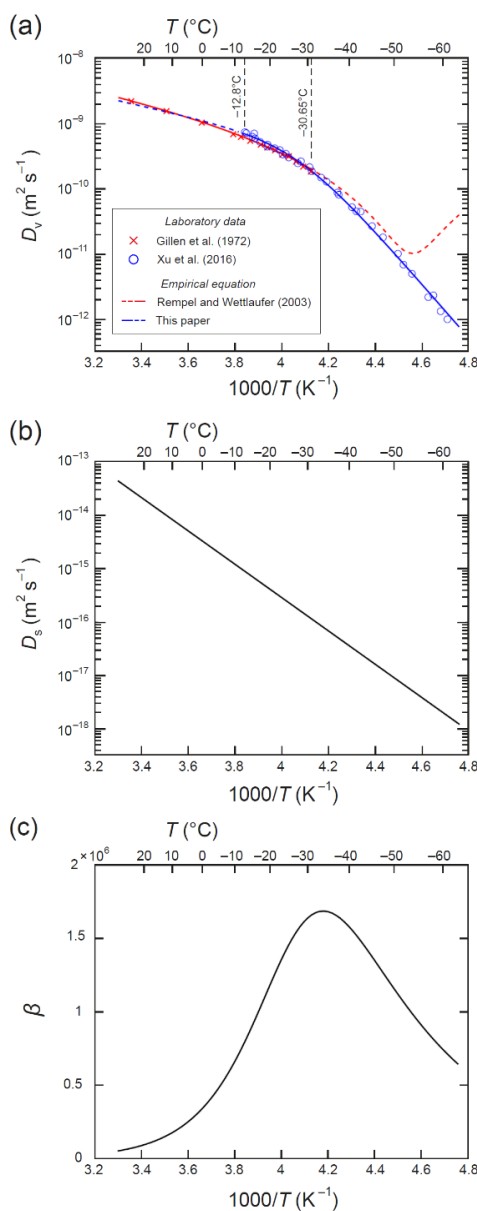

Figure 2. Isotopic diffusivities and their temperature dependence. (a) Arrhenius plot of the self-diffusivity of water, $D_v$. Symbols plot values based on laboratory measurements. Blue curve: our composite exponential in Eq. (A1), fitted to the data of Xu et al. (2016) and used to calculate $D_v$ in this paper. Red curve: quadratic fit by Rempel and Wettlaufer (2003) to the data of Gillen et al. (1972); the dashed portion evaluates their quadratic at $T$ below –31 °C, outside its region of applicability. (b) Self-diffusivity of monocrystalline ice $D_s$, calculated with Ramseier's (1967) empirical formula in Eq. (A2). (c) The liquid-to-solid diffusivity contrast $\beta$ (= $D_v/D_s$ – 1).





$$f = 1 + \frac{k_r^2}{k_z^2} \tag{9}$$

measures how much faster signals decay in the presence of veins, or equivalently, how much the veins increase the effective diffusivity of the system above $D_s$. Owing to their short-circuiting effect, excess diffusion operates ($f > 1$) even when $w = 0$. Our main interest is how $f$ varies with $w$. Note that $f$, $k_r$, $\zeta_R$ and $\zeta_I$ are functions of $k_z$.

Hitherto we seem to be mostly retracing the steps of Nye (1998) and Rempel and Wettlaufer (2003). But a key difference herein – and what distinguishes our findings – is that the decay-rate parameter $\zeta$ and the amplitude function $F$ in Eq. (7) are complex numbers when $w \neq 0$, since the problem is then no longer symmetric in $z$. Particularly, a non-zero $\zeta_I$ implies signal migration at the velocity $v = \zeta_I D_s / k_z$, and we anticipate $F(r) = F_R(r) + iF_I(r)$, with the signal phase given by $\theta(r) = \tan^{-1}(F_I / F_R)$, varying with radius under the advection. Symmetry considerations for how the system behaves when the vein-water flow

direction is reversed predict $\zeta_R$ (hence $f$) and $\zeta_I$ (hence $v$) to be even and odd functions of $w$, respectively.

        Now, substituting (7) into (3), (4) and (5) leads to the Bessel Equation

$$F'' + \frac{F'}{r} + (k_r^2 + i\zeta_I)F = 0 , \tag{10}$$

with the boundary conditions

$$F'(b) = 0 , \tag{11}$$

$$F''(a) + \frac{1-2\alpha}{a} F'(a) + \left( \beta k_z^2 + \frac{ik_z w}{D_s} \right) F(a) = 0 . \tag{12}$$

Suppose $k_r^2 + i\zeta_I \equiv s^2 = (s_R + is_I)^2$ such that

$$k_r^2 = s_R^2 - s_I^2 \quad \text{and} \quad \zeta_I = 2s_R s_I. \tag{13}$$

Then, in terms of Bessel functions, the analytic solution of Eqs. (10) to (12) is

$$F(r) = J_0(sr) - \frac{J_1(sb)}{Y_1(sb)} Y_0(sr) \tag{14}$$

(or any constant multiples), in which $s$ at each wavenumber $k_z$ satisfies

$$\frac{2\alpha s}{a\left[s^2 - \beta k_z^2 - ik_z w/D_s\right]} - \frac{J_0(sa)Y_1(sb) - Y_0(sa)J_1(sb)}{J_1(sa)Y_1(sb) - Y_1(sa)J_1(sb)} = 0 . \tag{15}$$

These results are equivalent to those of Rempel and Wettlaufer (2003) when $\alpha = 1$ and $w = 0$. When $w \neq 0$, $s$ is complex, and





the Bessel functions take complex values.[2] For each wavelength $\lambda$ (or $k_z$), we solve for $s$ numerically by Newton's Method, taking the left side of (15) as the function whose root is sought. We then compute $k_r$, $\zeta_R$, $\zeta_I$, $f$ and $F(r)$ via (13), (8), (9) and

(14). The numerical code of these solution steps is given at the online repository of the paper.

Ice-core measurements of $\delta$ are often made on horizontal layers spanning multiple grains, so it is useful to consider the mean value of $\delta$ at each depth in the model (Fig. 1):

$$
\begin{aligned}
\bar{\delta}(z,t) &= \frac{1}{\pi b^2}\left(\int_a^b 2\pi r\delta\ dr + \pi a^2\ \delta|_{r=a}\right) \\
&\approx \delta_0 + \frac{2\delta_1}{b^2}\exp[-D_s\zeta t + ik_z z]\int_a^b rF(r)\ dr\ .
\end{aligned}
\tag{16}
$$

This expression shows that the section-mean signal at each wavelength is itself sinusoidal, with the same decay rate, decay-

rate enhancement factor and migration velocity as for the component signals at different radii. The approximation (based on $a \ll b$) recognises a negligible contribution to the mean signal from the vein water.

## 3    Results and analysis: excess diffusion at the grain scale

We proceed to analyse computed results to understand the impact of vein-water flow on signal evolution. Notably, we show that advection perturbs $\delta$ in such a way that amplifies the short-circuiting to accelerate signal smoothing, raising $f$ above the

enhancement factor of Rempel and Wettlaufer (2003). Through successive numerical experiments, we elucidate the mechanism and key controls on $f$. We explain relevant properties of the model along the way.

Our experiments here explore signal wavelengths $\lambda$ in the range 0.001–0.15 m and vein-flow velocities $w$ up to $\sim 10^2$ m

yr$^{-1}$, for $T = -32\ °C$ or $-52\ °C$. These temperatures resemble those measured in the upper ice column at ice-core sites in central Greenland and central Antarctica, respectively (e.g., Fig. 8). The higher temperature is close to 241 K, which Rempel and

Wettlaufer (2003) chose based on the GRIP site conditions for their calculations. The qualitative dependence of $f$ on the vein and grain radii found by these authors ($f$ increases with $a$ and decreases with $b$) is unchanged in our model, and is not the focus of our study, so we assume constant radii $a = 1\ \mu m$ and $b = 1\ mm$ in the experiments. We assume $\alpha \equiv 1$, the approximation used by Rempel and Wettlaufer (2003), so the results are applicable for either $\delta^{18}O$ or $\delta D$.

---

[2] One hopes to express the solution of Eq. (10) in terms of real transcendental functions, as in the case $k_r > 0$, $\zeta_I \equiv 0$ (which gives Bessel functions with real arguments) or the case $k_r \equiv 0$, $\zeta_I < 0$ (which gives Kelvin functions; see Abramowitz and Stegun (1972)). For $k_r$ and $\zeta_I$ both non-zero, however, we have not found such functions in the literature and need to evaluate the Bessel functions in Eqs. (14) and (15) for complex arguments. This is done straightforwardly in MATLAB.





Our range for $w$ is informed by the theoretical estimates of Nye and Frank (1973) for glaciostatically-driven water flow

through a vein network, listed in their Table 2. We bear in mind that their highest vein-flow velocity (900 m yr$^{-1}$) assumes a

high liquid fraction or porosity ($\sim 10^{-3}$) that is probably uncommon for polar ice, so we explore up to values of $w$ an order of

magnitude smaller. In the Discussion we comment more on the lack of observations of vein-water flow.

It is noteworthy that Rempel and Wettlaufer (2003) ignored isotope advection by vein-water flow on the basis that the

Peclet number Pe $= w/D_\mathrm{v}k_z$ is small for the signal wavelengths of interest ($\sim$ dm or less for annual layers). Pe measures the

ratio of the advection (fourth) term to the diffusion (third) term in Eq. (5). It is indeed small in our experiments. But as will be

seen shortly, although the $\delta$-field is modified only slightly when $w \neq 0$, this change can increase $f$ significantly.

Figure 3 shows the computed pattern of $\delta$ in the ice annulus, in three experiments with $w = 0$, 5 and 50 m yr$^{-1}$, at $\lambda = 2$

cm and $T = -32°$C. All three signals decay with time; those in Fig. 3b and 3c migrate downward at constant velocity. We focus

on examining the spatial part of the solution in Eq. (7) – the colour maps plot Re[$F(r)\exp(ik_zz)$] (or $|F(r)|\cos(k_zz + \theta(r))$ where


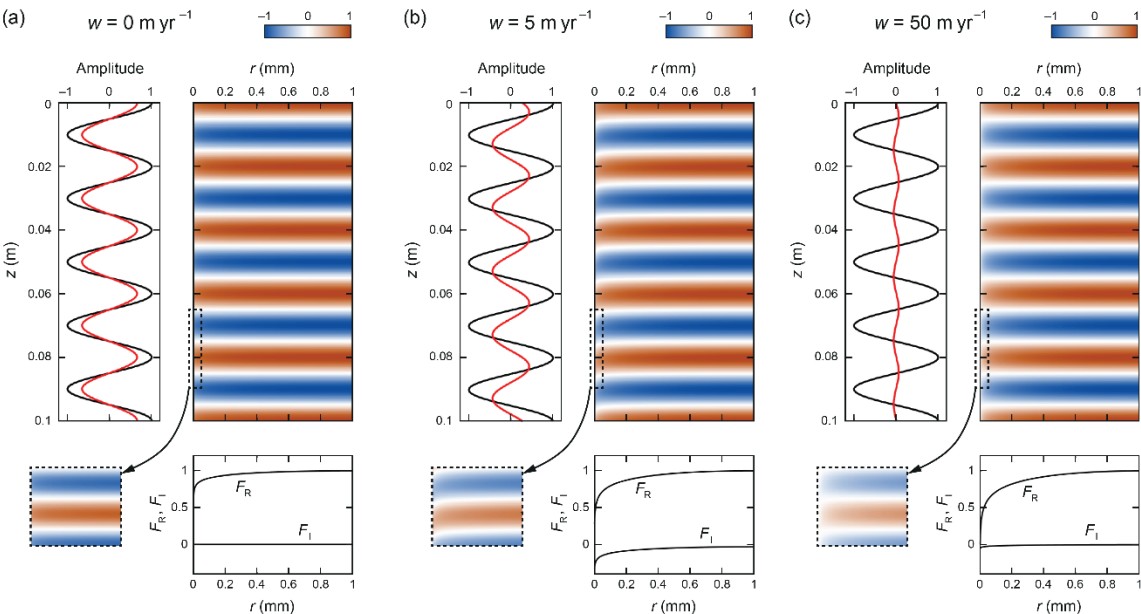

Figure 3. Patterns of $\delta$ calculated with Eqs. (7) to (15) for $\lambda = 0.02$ m, $T = -32$ °C, and $w$ equal to: (a) 0, (b) 5, and (c) 50 m yr$^{-1}$. The

enhancement factors in these experiments are $f = 2.11$, 3.24 and 4.26, respectively. Each colour map samples a radial cross-section of the

three-dimensional ice annulus in Fig. 1 and is shown with a horizontal exaggeration of 50. Dashed boxes expand on the details near $r = 0$.

The vein boundary lies at $r = a = 1$ $\mu$m. The panel under each map plots the real and imaginary parts of the amplitude function $F(r)$. The

panel left of each map plots the $\delta$-variations at the vein (red) and in the farthest part of the grain interior ($r = b$, black).





$\theta$ is the signal phase defined earlier) without the time element. When plotting each pattern, we scale its amplitude such that

$|F(b)| = 1$. This choice facilitates comparison of the $\delta$-variations along the vein with those at $r = b$.

Nye's short-circuiting occurs when $w = 0$ (Fig. 3a), as expected. As we move from the grain interior towards the vein,

the sinusoidal signals decrease in amplitude sharply following $F(r)$, very close to the vein (dashed box, Fig. 3a). Consequently, isotopes diffuse from the grain interior towards the vein along the peak ridges of the signal, and in the opposite direction along troughs. This pattern of transverse isotopic exchange between vein and ice is driven by fast diffusion along the vein smoothing the signal there and is what causes the entire signal to decay faster than the baseline decay rate, $D_s k_z^2$ (as governed only by isotope diffusion vertically between the ridges and troughs). The enhancement factor in this experiment is $f = 2.18$, as calculated

by Rempel and Wettlaufer (2003).

When we switch on vein-water flow (Figs. 3b, c), $f$ is amplified by a novel effect. Advection shifts the vein signal down relative to the interior, inducing a sheared pattern of $\delta$-variations in a thin layer next to the vein boundary, of negative phase ($F_I < 0$). At $w = 5$ m yr$^{-1}$ (Fig. 3b), the sinusoidal variations in the layer have a 'tail-like' appearance in colour. Because their phase shift increases towards the vein, there are high lateral gradients in $\delta$ at the vein end of the signal peaks and troughs, and

they drive a stronger diffusive isotopic exchange between vein and ice (than in the no-flow case) which accelerates the signal decay: $f = 3.29$ in this case. The profile of $F_R(r)$ and signal amplitude at $r = a$ are correspondingly reduced.

When $w$ is raised to 50 m yr$^{-1}$ (Fig. 3c), the shear layer becomes 'sheet-like'. Strong advection causes $\delta$ outside the vein to interact with $\delta$ in the vein much higher up, and the coupling of this with diffusion in the ice diminishes the isotopic variations along and immediately outside the vein to near zero. One can think of the pattern in the layer now as due to the tails of high

(/low) $\delta$ extending far down to cover the next trough (/ridge), with $\delta$ averaging out sideways by diffusion. Compared to the last experiment, the $F_R$-profile is drawn down even more, the transverse isotopic exchange still stronger. The enhancement $f = 4.26$ is nearly maximised, as the signal pattern is close to what it would be at the $w \to \infty$ limit, with no variations along the vein, as in the Nye model (we confirmed this in experiments that took $w$ above 50 m yr$^{-1}$).

The last finding implies that at any signal wavelength, the high flow limit ($w \to \infty$) yields the same enhancement as the

Nye model limit ($\beta \to \infty$). This equivalence arises because in Eq. (5) $w \to \infty$ drives $\partial\delta/\partial z|_{r=a}$ to zero, whereas $\beta \to \infty$ drives

$\partial^2\delta/\partial z^2|_{r=a}$ to zero, and both yield the constant vein boundary condition $\delta \equiv \delta_0$ to precondition the same isotopic pattern. It

follows that $f$ at high flow in our model asymptotically reaches Nye's enhancement factor, and since $f$ is minimum at $w = 0$

(and equal to Rempel and Wettlaufer's enhancement factor), $f$ must take an intermediate value in $0 < w < \infty$.





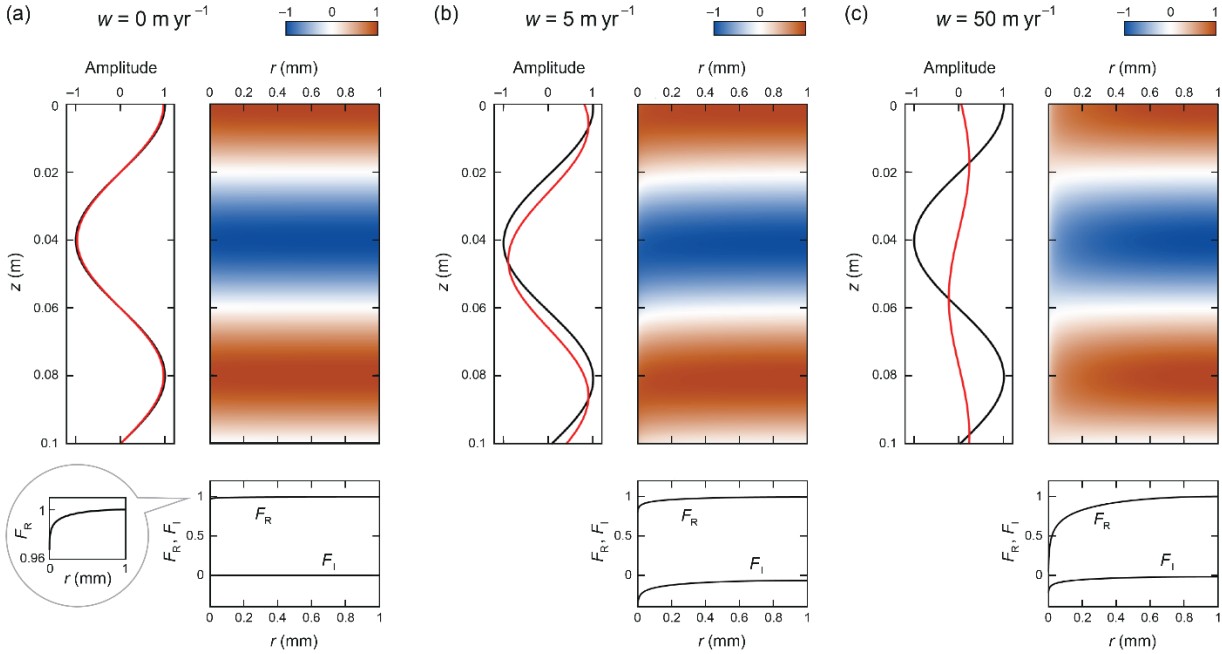

Figure 4. Patterns of $\delta$ computed for $\lambda = 0.08$ m, $T = -32$ °C, and $w$ equal to (a) 0, (b) 5, and (c) 50 m yr$^{-1}$. The enhancement factors in these experiments are $f = 2.63$, 9.01 and 50.2, respectively. The figure has a similar layout as Fig. 3. Inset in (a) expands on the variations of $F_R$.

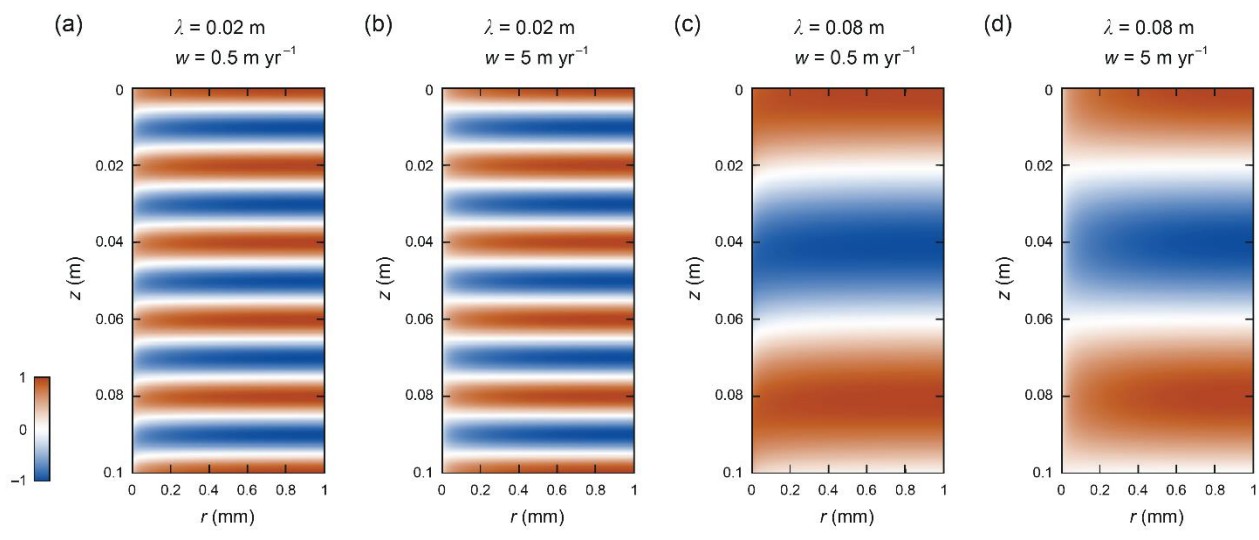

Figure 5. Patterns of $\delta$ computed for $T = -52$ °C, at (a) $\lambda = 0.02$ m, $w = 0.5$ m yr$^{-1}$, (b) $\lambda = 0.02$ m, $w = 5$ m yr$^{-1}$, (c) $\lambda = 0.08$ m, $w = 0.5$ m yr$^{-1}$, and (d) $\lambda = 0.08$ m, $w = 5$ m yr$^{-1}$. The enhancement factors are $f = 3.68$, 4.27, 14.7 and 51.9, respectively.





Next we consider wavelength control. At fixed $w$, longer signals experience a higher decay-rate enhancement (but remember their baseline decay rate is lower). Figure 4 demonstrates this with a modified set of experiments, for $\lambda = 8$ cm, at

the same temperature and flow velocities as before. The shear-layer mechanism again operates when $w > 0$. The tails lengthen as $w$ is increased, although the pattern is still in the 'tail regime' at 50 m yr$^{-1}$; at this longer wavelength, a higher $w$ is needed to shift the variations far enough for transition to the 'sheet regime'. In all three cases, $f$ is higher (respectively $\approx 1.3$, 3 and 12 times greater) than before. The reason lies in the relative contribution of (i) vertical diffusion between signal ridges and troughs and (ii) lateral diffusive exchange between vein and ice, in driving the signal decay. For signals that are short compared to the

grain radius ($\lambda \ll b$), vertical diffusion dominates over lateral exchange, so vein-water flow increases $f$ minimally via the shear-layer mechanism. For long signals ($\lambda \gg b$), the lateral exchange is more significant, so the shear-layer mechanism amplifies $f$ more strongly. Note that all three $F_R$-profiles in Fig. 4 curve down less than those in Fig. 3, but the attendant reduced lateral exchange rates are still higher than the vertical diffusion rates, which are 16 times less at $\lambda = 8$ cm than at $\lambda = 2$ cm. Rempel and Wettlaufer (2003) made similar arguments to explain the wavelength control on $f$ in the system without vein flow. Here

we have added the shear-layer mechanism to the considerations.

Results for colder ice ($T = -52$ °C, Fig. 5) predict higher enhancements at the same values of $\lambda$ and $w$, and shear-layer transitions at lower vein-flow velocities. For both the 2 cm and 8 cm signals, the tail-to-sheet transition is now largely complete when $w$ reaches 5 m yr$^{-1}$ (Fig. 5; cf. Figs. 3 and 4). These changes are not due mainly to the change in diffusivity contrast $\beta$ (= $D_v/D_s - 1$), but rather to the reduced $D_s$ at low temperature (Fig. 2), which raises the importance of vein-flow assisted lateral isotope exchange compared to vertical diffusion in the grains in smoothing the signal. We study the temperature control more

below, where it will be seen that the dominance of these factors $D_s$ and $\beta$ reverses at low vein-flow velocities.

The mechanism detailed here – initiation of the shear layer by vein-water flow, its progression through the tail and sheet regimes as the magnitude of $w$ is increased, and how the layer isotopic gradients amplify the short-circuiting to accelerate signal decay – is universal across our experiments. To help readers visualise the evolution, we show in Movies S1 and S2

continuous versions of Figs. 3 and 4, with $w$ changing in small steps, covering upward as well as downward water flow.

Having explored the spatial interactions behind the decay-rate enhancement amplification, we report the influences of wavelength and vein-flow velocity more comprehensively by computing curves of $f(\lambda)$ at fixed $w$ (Fig. 6) and surfaces of $f$ over the $\lambda$–$w$ parameter space (Fig. 7). We do this for $T = -32$ °C and $T = -52$ °C, plotting $\log_{10} f$ and the signal migration velocity $v$ also. Figures 6 and 7 confirm that $f$ increases monotonically with $\lambda$ and $|w|$, and, at each $\lambda$, $f$ increases from Rempel

and Wettlaufer's $f$ value at $w = 0$ towards a maximum (Nye's enhancement factor) as $|w| \to \infty$. Importantly, while for centi-

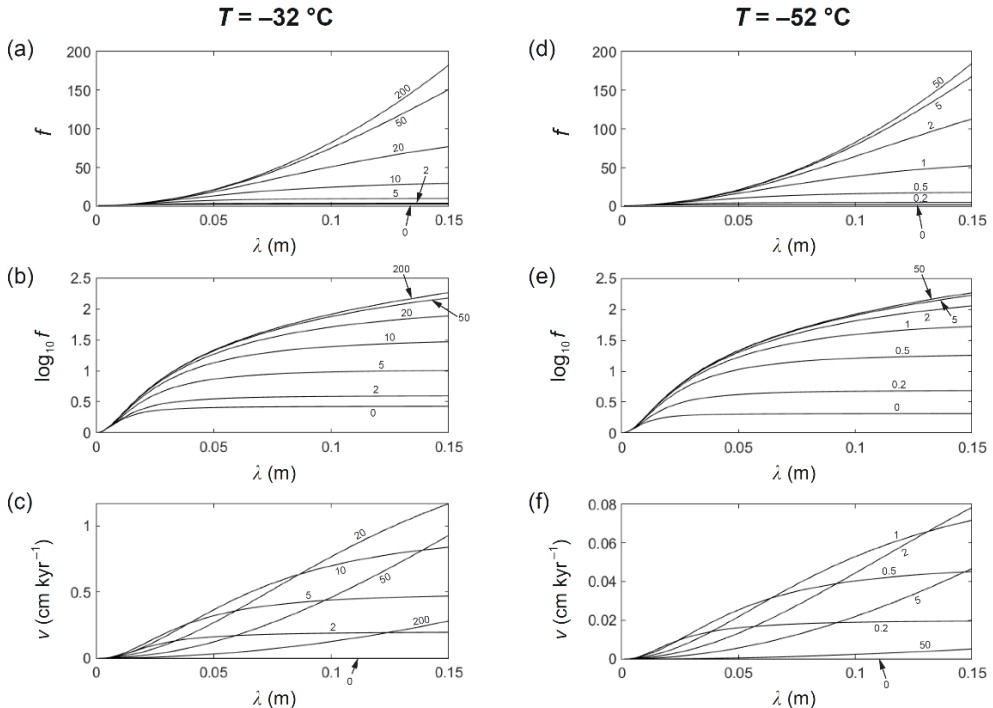

Figure 6. Computed curves of signal decay-rate enhancement factor $f$, $\log_{10} f$ and signal migration velocity $v$ versus signal wavelength $\lambda$, at (a–c) $T = -32$ °C and (d–f) $T = -52$ °C, for different vein-flow velocities $w$ (labels by the curves, in m yr$^{-1}$). The lowest curves in $f$ and $\log_{10} f$ portray Rempel and Wettlaufer's (2003) enhancement factor. The highest curves approach Nye's (1998) enhancement factor.

metre to decimetre-scale signals $f$ is a few times without vein-water flow, it increases to $\sim 10^1$–$10^2$ with vein-water flow. The increase is steepest at $|w| \sim 10$ to 20 m yr$^{-1}$ at –32 °C and $|w| \sim 1$ m yr$^{-1}$ at –52 °C. Accordingly, for the upper parts of ice cores from central Greenland, West Antarctica and coastal Antarctica, where $T \approx -20$ °C to –30 °C is common, the extra enhancement above Rempel and Wettlaufer's $f$ is limited until $w$ exceeds a few metres per year (Fig. 6a–b). For the upper parts of ice cores in central East Antarctica, such as at the EPICA Dome C, Dome Fuji and Vostok ice-core sites, where $T \approx -50$ °C, the extra enhancement is already significant at $w \sim 0.5$ m yr$^{-1}$ (Fig. 6d–e). Results computed using the fractionation coefficients for oxygen and deuterium instead of $\alpha = 1$ (Figs. S1 and S2) differ minimally from those in Figs. 6 and 7.

The surfaces of $f$ at the two temperatures have similar shape but different scales in $w$ (Fig. 7). The surface for –52 °C is in fact almost exactly a compressed version in the $w$-direction of the surface for –32 °C. Movie S3 illustrates this "compressional scaling" as $T$ varies from –20 °C to –60 °C. The surface always approaches the same profile $f(\lambda)$ as $|w| \to \infty$ (also see Figs. 6a, d) to yield Nye's enhancement factor, which does not depend on temperature, because $D_v$ and $D_s$ do not

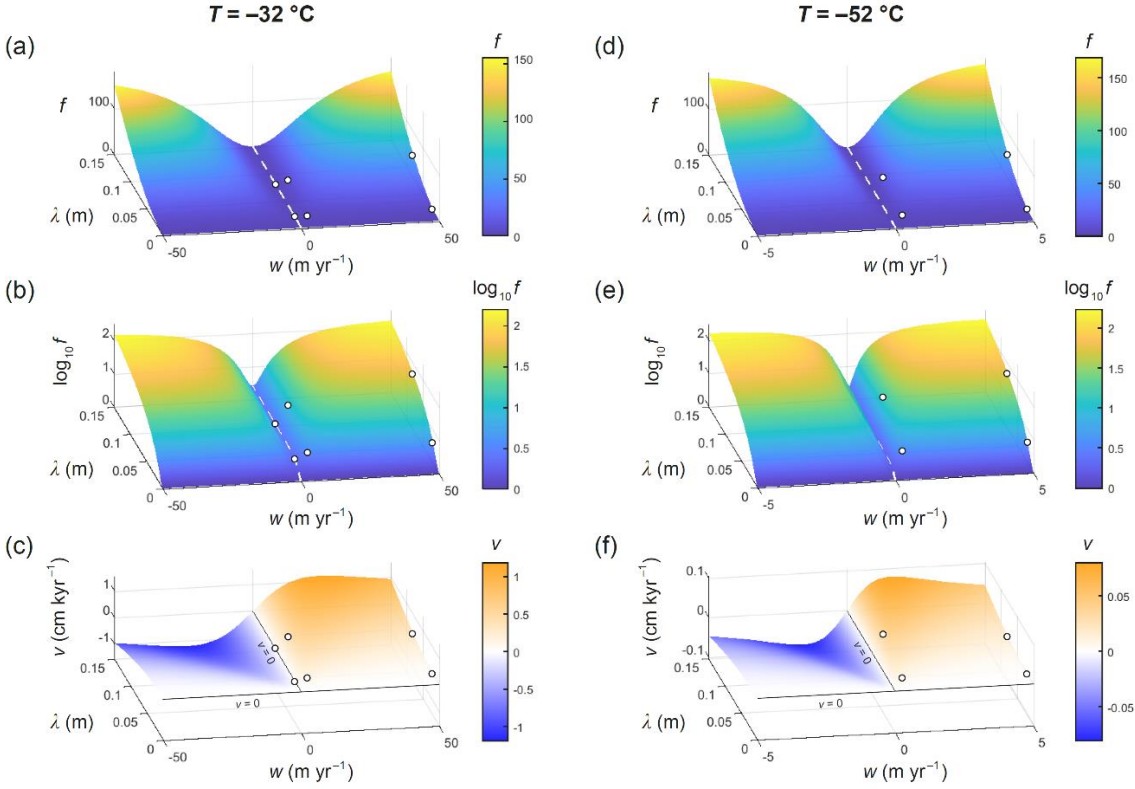

Figure 7. Computed signal decay-rate enhancement factor $f$, $\log_{10}f$ and signal migration velocity $v$ over the $\lambda$–$w$ parameter space at: (a–c) –32 °C and (d–f) –52 °C. White dashed curves delineate $f(\lambda)$ at $w = 0$, which is the enhancement factor in Rempel and Wettlaufer's (2003) model. Circles locate the experiments of Figs. 3, 4 and 5. The curves in Fig. 6 are transects of these surfaces at fixed $w$.

enter the model to influence $\zeta$ when $w \to \infty$ or $\beta \to \infty$ (Eqs. (3) to (5)) (remember the baseline decay rate does increase with

$T$ via $D_s$). However, the surface evolves not merely by compressional scaling. A subtle change also occurs in the valley near

$w \approx 0$: there, $f$ at –32 °C exceeds $f$ at –52 °C slightly (the lowest curves in Figs. 6b and 6e). Consequently, as $T$ is reduced, the

surface contracts towards $w = 0$, causing $f$ to rise at moderate to large values of $|w|$ (yielding the earlier result that $f$ decreases

with temperature) but to drop near $w \approx 0$ (this is too small to be visible in Movie S3).

These temperature controls can be explained by a model scaling analysis. As detailed in Appendix B, with constant vein

and grain radii ($a$ and $b$ fixed), three dimensionless parameters govern the signal pattern in the ice annulus and the associated

signal decay rate: (i) the ratio of the signal wavelength to the grain radius, $\lambda/b$; (ii) the ratio of isotope advection by vein-water

flow to isotope diffusion in the ice, $wb/D_s$; and (iii) the diffusivity contrast $\beta$. (The Peclet number considered by Rempel and

Wettlaufer (2003) is a combination of these parameters.) It follows that the enhancement factor $f$ has the functional form $f(\lambda/b,$





$wb/D_s$, $\beta$), whose shape is portrayed by the surfaces in Fig. 7. The influences of $\lambda$ and $w$ in the first two arguments of this function were explored in earlier experiments. A temperature change affects $f$ via both its second and third arguments, because

$D_s$ and $\beta$ vary with $T$ (Fig. 2). The compressional scaling stems from the second argument, $wb/D_s$. In contrast, the influence on $f$ by the third argument $\beta$ (over its range of interest, $\sim 10^6$) is weak, and prevails only when $f$ is small – near $w \approx 0$. There, $f$ decreases when $T$ is reduced from –32 °C to –52 °C, because a decrease in $\beta$ (Fig. 2c) weakens the short-circuiting.

Turning to the migration velocity $v$ (Fig. 6c, f; Fig. 7c, f), the model predicts signals to move in the direction of vein-water flow, at speeds that reach a maximum at intermediate $w$, and are higher for long signals and at high temperature, of up

to $\sim 1$ cm kyr$^{-1}$. Most speeds on the parameter space are much lower. Hence signal migration is slow, in the sense that long (at least millennial) timescales are needed to displace centimetre and decimetre-scale annual signals against the ice and other ice-core proxies by a wavelength or more. The relative inaccuracy caused by this on the age scales determined by the counting of $\delta$-cycles on isotope records is negligible. Compressional scaling applies also to $v$ (Fig. 7, Movie S3), which has the form $(D_s/b)g(\lambda/b, wb/D_s, \beta)$ where the function $g$ differs from $f$ (Appendix B). The prefactor $D_s/b$ explains why migration slows as

temperature is reduced.

In their calculations, Rempel and Wettlaufer (2003) and Johnsen et al. (2000) accounted for the misorientation of veins from the vertical in the three-dimensional vein network, by reducing $D_v$ by a bulk tortuosity factor $\tau_f = 3$. Doing this in our experiments would lower $\beta$ by $\approx 3$ times and alter their results numerically, but not change our qualitative findings.

## 4  Implications for diffusion-length studies

To explore how much excess diffusion modulated by vein-water flow impacts signal smoothing down the ice column, we simulate diffusion-length profiles for ice-core sites in Greenland and Antarctica, in a forward model testing $w$. We compare the results against profiles modelled without excess diffusion, and query whether they match the level of excess diffusion inferred from isotope records. We also consider the impact on diffusion-length-based temperature reconstructions.

We use the well-established theory of Johnsen (1977) for these calculations, treating what happens below the firn only.

In a moving coordinate system where $z$ measures depth below a material horizon in the ice as it descends towards the bed, isotopic signals evolve according to

$$\frac{\partial \overline{\delta}}{\partial t} = D(t)\frac{\partial^2 \overline{\delta}}{\partial z^2} - \dot{\varepsilon}_z(t)z\frac{\partial \overline{\delta}}{\partial z}, \tag{17}$$

where $t$ is the age of the horizon, $\dot{\varepsilon}_z$ ($< 0$) is the local vertical strain rate, and $\overline{\delta}$ is the section-mean signal in Eq. (16). Since





the enhancement factor $f$ applies to this signal (Sect. 2), the bulk-ice isotopic diffusivity in (17) is given by

$D(t) = D_s(T)f(\lambda, w) ,$                                                                 (18)

where the dependence on age arises through $T$, $\lambda$ and other controls of $f$ that vary as the signal descends ($\beta$, potentially also $w$,

$a$ and $b$). We model a steady-state ice column with fixed temperature and ice velocity profiles. Given the ice thickness, the

surface accumulation rate and strain-rate profile, the age–depth scale is determined, and signals with the wavelength $\lambda_0$ at the

firn transition shorten to the wavelength $\lambda = \lambda_0 S(t)/S(t_0)$ at age $t$, where $S(t) = \exp \int_0^t \dot{\varepsilon}_z(\tau) \, \mathrm{d}\tau \leq 1$ is the thinning function,

and $t_0$ is the firn transition age. The normalisation of $S$ by $S(t_0)$, absent in studies that track signals from the ice-sheet surface,

accounts for the minor thinning that has taken place by $t = t_0$.

        According to Johnsen's (1977) solution of Eq. (17), one can track the amplitudes of different harmonics (Fourier

components) of the signal separately[3], by using the diffusion length $\sigma$, which measures the root mean square distance travelled

by diffusing isotopes. Specifically, the squared diffusion length $\sigma^2$ obeys the differential equation

$\dfrac{\mathrm{d}\sigma^2}{\mathrm{d}t} - 2\dot{\varepsilon}_z(t)\sigma^2 = 2D(t) ,$                          (19)

and each harmonic attenuates by the ratio $R = \exp(-2\pi^2\sigma^2/\lambda^2)$ as $\sigma$ and $\lambda$ evolve down-column. It follows that a harmonic is

attenuated strongly when its wavelength shortens to less than $\sigma$ (e.g., Gkinis et al., 2021). In our simulations, we specify an

initial value $\sigma = \sigma_{\mathrm{firn}}$ at the firn transition – taken from studies of firn isotope diffusion – to circumvent the need to model firn

processes. The ratio tracking signal amplitude in the ice is then

$R_i = \exp\left[-2\pi^2(\sigma^2 / \lambda^2 - \sigma_{\mathrm{firn}}^2 / \lambda_0^2)\right].$                          (20)

        Following Gkinis et al. (2014), we decompose $\sigma^2$ into a part due to isotopic diffusion in ice, $\sigma_{\mathrm{ice}}^2$, and another part

inherited from firn isotopic diffusion that thins under the compression; thus,

$\sigma^2(t) = \sigma_{\mathrm{ice}}^2(t) + \sigma_{\mathrm{firn}}^2 \left[\dfrac{S(t)}{S(t_0)}\right]^2 .$                          (21)

---

[3] This is because their wavelengths follow different histories of shortening. To see this, notice Eq. (17) has the characteristic velocity
$\mathrm{d}z/\mathrm{d}t = \dot{\varepsilon}_z z$ , which motivates a change of the depth variable to $Z = z/S(t)$. Changing the time variable also via $\tau = \int_0^t D(t)/S(t)^2 \, \mathrm{d}t$
converts (17) to the classic diffusion equation $\partial\delta/\partial\tau = \partial^2\delta/\partial Z^2$, whose Fourier components evolve independently.





Substituting for $\sigma^2$ in (19), using (18) for $D$, yields

$$\frac{\mathrm{d}\sigma_{\mathrm{ice}}^2}{\mathrm{d}t} - 2\dot{\varepsilon}_z \sigma_{\mathrm{ice}}^2 = 2D_s f(\lambda, w), \qquad (22)$$

with $\sigma_{\mathrm{ice}} = 0$ at $t = t_0$. This equation is straightforward to solve analytically by an integrating factor (Gkinis et al., 2014) or

numerically by the finite-difference method (as done in our simulations below).

In reconstructions of firn temperature (e.g., Gkinis et al., 2014; Holme et al., 2018), Eq. (21) is rearranged as

$$\sigma_{\mathrm{firn}}^2 = \frac{\sigma_{\mathrm{obs}}^2 - \sigma_{\mathrm{ice}}^2}{[S/S(t_0)]^2}, \qquad (23)$$

to allow $\sigma_{\mathrm{firn}}$ at a given age to be found from (i) the thinning $S$, (ii) the modelled value of $\sigma_{\mathrm{ice}}$ (from (22)) and (iii) the diffusion

length $\sigma_{\mathrm{obs}}$ estimated from isotopic signals in the ice core – at the same age. Firn isotope diffusion modelling is then used to

invert $\sigma_{\mathrm{firn}}$ for temperature. The estimation of $\sigma_{\mathrm{obs}}$ involves fitting $P_0(k) = P_0 R^2 = P_0 \exp(-k^2\sigma^2)$ ($k = 1/\lambda$ is the wavenumber) to

the power spectral density (PSD) graph of the measured signals, assuming a white-noise input signal at the surface, i.e.,

constant $P_0$; see Kahle et al. (2018) for different approaches to this estimation.

Eq. (22) reveals a notable consequence of excess diffusion ($f > 1$) for the diffusion-length estimation. Besides raising $\sigma_{\mathrm{ice}}$

(thus $\sigma$) to accelerate signal decay, excess diffusion makes $\sigma_{\mathrm{ice}}$ *wavelength dependent*: this does not arise if the bulk ice has

Ramseier's diffusivity ($f \equiv 1$), as assumed in most past studies. With excess diffusion, $\sigma_{\mathrm{ice}}$ varies with $\lambda$ and the initial wave-

length $\lambda_0$, so the harmonic components have different diffusion-length histories. Their spectral power now decays as $\exp(-$

$k^2\sigma^2)$, where $\sigma$ decreases with $k$ (rather than being constant), as $f$ increases with $\lambda$ (Sect. 3). Since $f = 1$ at zero $\lambda$ only (Figs. 6

and 7), $\sigma$ exceeds the $\sigma$-value for monocrystalline ice at all $k < \infty$. Fitting of the resulting non-parabolic PSD thus overestimates

$\sigma_{\mathrm{obs}}$, in the context of firn-temperature reconstructions assuming Ramseier's diffusivity for the ice in (22) and (23).

More precisely, our model implies that fitting $\exp(-k^2\sigma^2)$ to the PSD decay of the signal to find $\sigma$ is no longer appropriate

when excess diffusion operates; strictly speaking, the fit should be made with $\exp[-k^2(\sigma_{\mathrm{ice}}^2 + (\sigma_{\mathrm{firn}}S/S(t_0))^2)]$ to find $\sigma_{\mathrm{firn}}$, with

$\sigma_{\mathrm{ice}}$ (solution of (22)) varying with $k$ and $w$. We demonstrate the wavelength dependence of $\sigma_{\mathrm{ice}}$ in simulations below. The

dependence does not arise in the firn, because $\sigma_{\mathrm{firn}}$ is not a function of $\lambda$, according to models of firn isotope diffusion (Whillans

and Grootes, 1985; Johnsen et al., 2000, Gkinis et al., 2014, Gkinis et al., 2021). Note that when the diffusion length $\sigma$ develops

wavelength dependence, it refers to the root mean square displacement of isotopes only if the signal (at a given depth) has a

single wavelength, not if it is composed of multiple harmonics.





We proceed to examine diffusion-length profiles computed for ice-column conditions based on the GRIP site in

Greenland and the EPICA Dome C site in Antarctica (Fig. 8); for these sites we specify $\sigma_{\text{firn}}$ = 8 cm and 7 cm, respectively

(e.g. figure S2 of Gkinis et al. (2014)). Most of our model runs assume $a = 1$ μm and $b = 2$ mm, although some prescribe a

variable grain-radius profile from measurements (Fig. 8d, h). We set $\alpha \equiv 1$ again, and 65 m as the firn transition depth. The

age–depth scales yield $t_0 = 286.3$ a for GRIP and 2872.9 a for EPICA. At the firn transition, the wavelength of annual signals

is given approximately by the ice-equivalent surface accumulation rate: 0.23 m yr$^{-1}$ at GRIP and 0.023 m yr$^{-1}$ at EPICA.

Figure 9a–d presents $\sigma$-profiles simulated at GRIP for the annual signal ($\lambda_0 = 0.23$ m yr$^{-1}$), alongside profiles of the

enhancement $f$, ice diffusion length $\sigma_{\text{ice}}$, and the ratio $R_i$ tracking signal amplitude. At each depth, $\sigma$ increases with $w$ via its

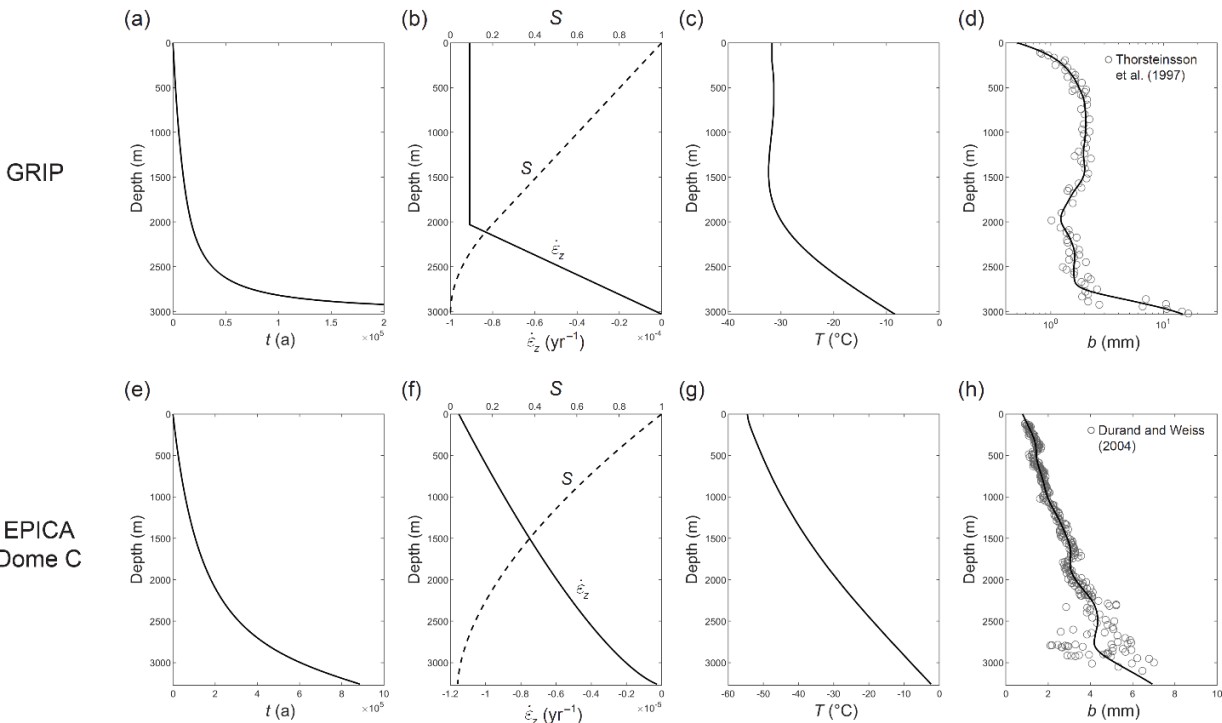

Figure 8. Depth profiles of glaciological variables used in our diffusion-length modelling for the (a–d) GRIP and (e–h) EPICA Dome C ice-
core sites. (a, e) Age–depth scale. (b, f) Strain rate $\dot{\varepsilon}_z$ and the associated thinning function $S$. (c, g) Ice temperature. (d, h) Grain-radius data
(circles) and spline curves used in our modelling. Ice flow at the GRIP site assumes the Dansgaard–Johnsen model with ice thickness $H =$
3029 m, kink height at 1000 m and surface accumulation rate $a_s = 0.23$ m yr$^{-1}$ ice equivalent. Ice flow at the EPICA site assumes the ice
submergence velocity $w_i = m_b + (a_s - m_b)(h/H)^{1.7}$, where $H = 3275$ m, $a_s = 0.023$ m yr$^{-1}$ ice equivalent, $h$ is height above the bed, and $m_b =$
0.0008 m yr$^{-1}$ is the basal melt rate. See Ng (2021) for additional details about these profiles.




modulation on $\sigma_{\text{ice}}$. Whereas excess diffusion with $w = 0$ (no vein-water flow) raises $\sigma$ and $\sigma_{\text{ice}}$ slightly above their values

based on Ramseier's diffusivity ($f = 1$), $w$ from several to tens of metres per year increases and modulates $\sigma_{\text{ice}}$ significantly,

with a strong impact on signal decay, down to $\approx 2300$ m depth. From about half way down the column, the amount of excess

diffusion diminishes rapidly with depth ($f \to 1$) due to severe shortening of the signal and increasing temperature (Sect. 3).

The $\sigma$ and $\sigma_{\text{ice}}$ profiles converge on the profiles for $f = 1$ near the base of the column because of this, and because the firn part

of $\sigma$ is vanishing, and because the long time spent by deep ice at similar strain rate and temperature allows the effects of

isotopic diffusion and vertical compression to balance, with $\sigma_{\text{ice}}$ equilibrating to $(-D_s(T)/\dot{\varepsilon}_z)^{1/2}$ in Eq. (22).

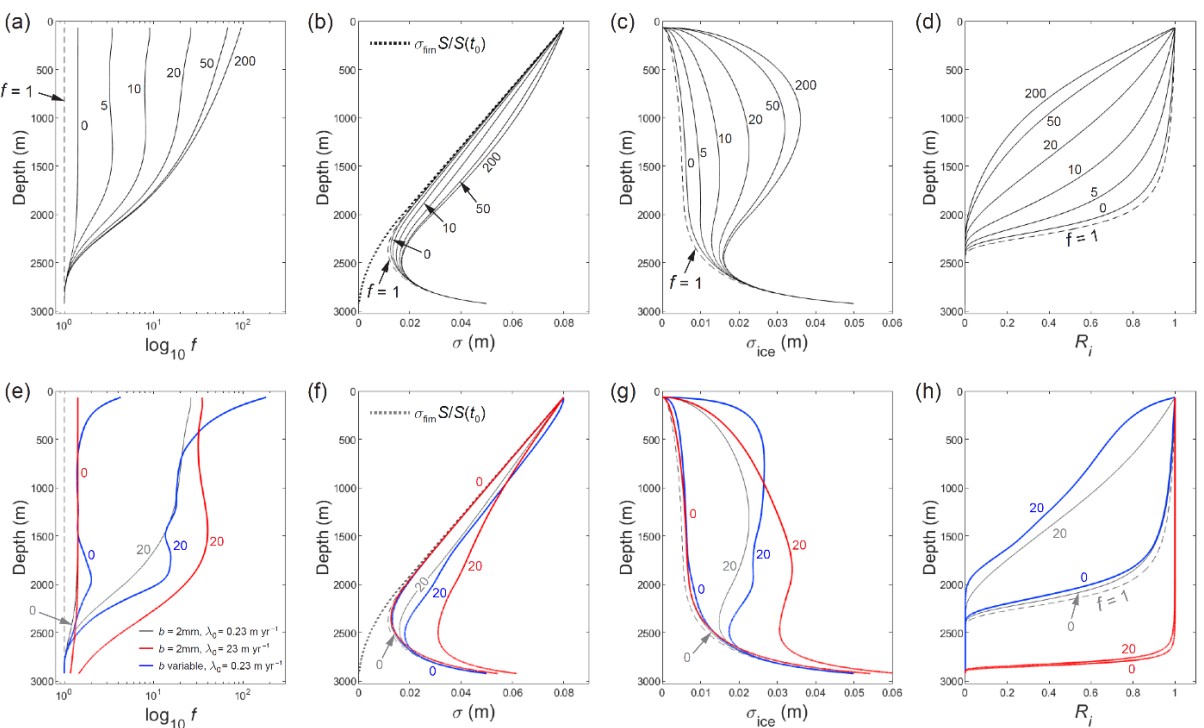

Figure 9. Computed depth profiles for the GRIP core site of (a, e) the enhancement factor $f$ (representing excess diffusion above the mono-

crystal diffusivity), (b, f) diffusion length $\sigma$, (c, g) ice diffusion length $\sigma_{\text{ice}}$, and (d, h) signal amplitude ratio $R_i$. Number by each curve

indicates the vein-water flow velocity $w$ in m yr$^{-1}$. In (a) to (d), all model runs study the annual signal ($\lambda_0 = 0.23$ m yr$^{-1}$), assuming the grain

radius $b = 2$ mm; dashed curves show results based on Ramseier's (1967) monocrystal diffusivity, i.e., $f = 1$; the dotted curve in (b) shows

the thinned firn diffusion length. In (e) to (h), blue curves report results based on the variable grain-radius profile in Fig. 8d; red curves report

results computed for signals with $\lambda_0$ a hundred times longer; grey curves show selected results from (a) to (d) for comparison.

The model run for $w = 20$ m yr$^{-1}$ in Fig. 9a–d mimics the diffusivity enhancement $f \sim 10$–30 found for annual signals in the GRIP Holocene ice by Johnsen et al. (1997) and Johnsen et al. (2000), predicting also the reduced excess diffusion in deeper ice (> 1600 m) dating to the Younger Dryas and the last glacial alluded in those studies. Although blockage of veins by the high dust content in stadial/glacial ice might explain this reduction (Johnsen et al., 1997; Johnsen et al., 2000), our model predicts a strong dependence of $f$ on the shortening signal wavelength when $w > 0$ (Fig. 6a, b) that provides an explanation. These considerations are unchanged if the run uses variable grain radius (Fig. 9e–h, blue curves for 20 m yr$^{-1}$). In contrast, the large enhancement $f \sim 10$–30 cannot be reproduced with $w = 0$ (Rempel and Wettlaufer's (2003) model) with constant or variable $b$ (Fig. 9e–h), not unless very large vein radii of $\sim 20$ to 200 $\mu$m are assumed.

The wavelength dependence of $\sigma$ and $\sigma_{\text{ice}}$ is apparent from two runs that study a signal with an initial wavelength 100 times longer than the annual signal (red curves in Fig. 9e–h; cf. grey curves). The dependence strengthens with $w$; we analyse its depth variations later. Even with strong excess diffusion (when $w = 20$ m yr$^{-1}$), the long signal survives much deeper than the annual signal (to $\approx 2800$ m; Fig. 9h) because its baseline decay rate is $10^4$ times lower.

Turning to the EPICA site, our interest is drawn to long signals, because annual signals are too short to survive isotopic diffusion in the firn (e.g., for $\lambda = 0.023$ m, the firn attenuation is $R = \exp(-2\pi^2 \sigma_{\text{firn}}^2 / \lambda^2) \sim 10^{-80}$). Figure 10a–c presents model runs for a millennial-scale signal with $\lambda_0 = 23$ m yr$^{-1}$. They show a similar modulation of the $f$ and $\sigma$-profiles by $w$ as seen in the GRIP runs; however, colder ice in the top half of the EPICA column than at GRIP (Fig. 8g, c) means that these profiles are more sensitive to $w$ (Sect. 3). The modulation at EPICA extends to nearer the base of the ice column because the lower strain rate (which shortens signals more slowly and slows the equilibration in (22)) causes $\sigma$ to approach the $f = 1$ curve slowly. The millennial signal survives into the deepest ice if $w \lesssim 30$ m yr$^{-1}$ (Fig. 10c).

What conditions at EPICA can produce the long diffusion lengths $\sigma \sim 40$ to 60 cm inferred for ice at the depth of MIS 19 ($\sim 3170$ m), and thus strong suppression of millennial signals and near-complete absence of sub-millennial signals there? Pol et al. (2010) surmised excess diffusion as necessary. Like the $\sigma$-profile they simulated, our result for $f = 1$ yields only $\sigma \approx 16$ cm at that depth (Fig. 10b). The runs with excess diffusion at constant $b$ show that $w \approx 80$ to 150 m yr$^{-1}$ generates enough excess diffusion, but the corresponding $\sigma$-profiles bulge at $\approx 1000$ m to attenuate the millennial signal strongly mid-column (Fig. 10b, c). Other ways of achieving $\sigma \sim 40$ to 60 cm in the deepest ice, with small or no bulge in $\sigma$ that allows a sizeable signal to survive past $\approx 2500$ m depth, but not to $\approx 3100$ m, are shown in Fig. 10d–h. They assume $w$-profiles ramping up towards the bed – linearly, parabolically, or linearly/nonlinearly near the base – that all require $w \gtrsim 150$ m yr$^{-1}$ in deep ice. If



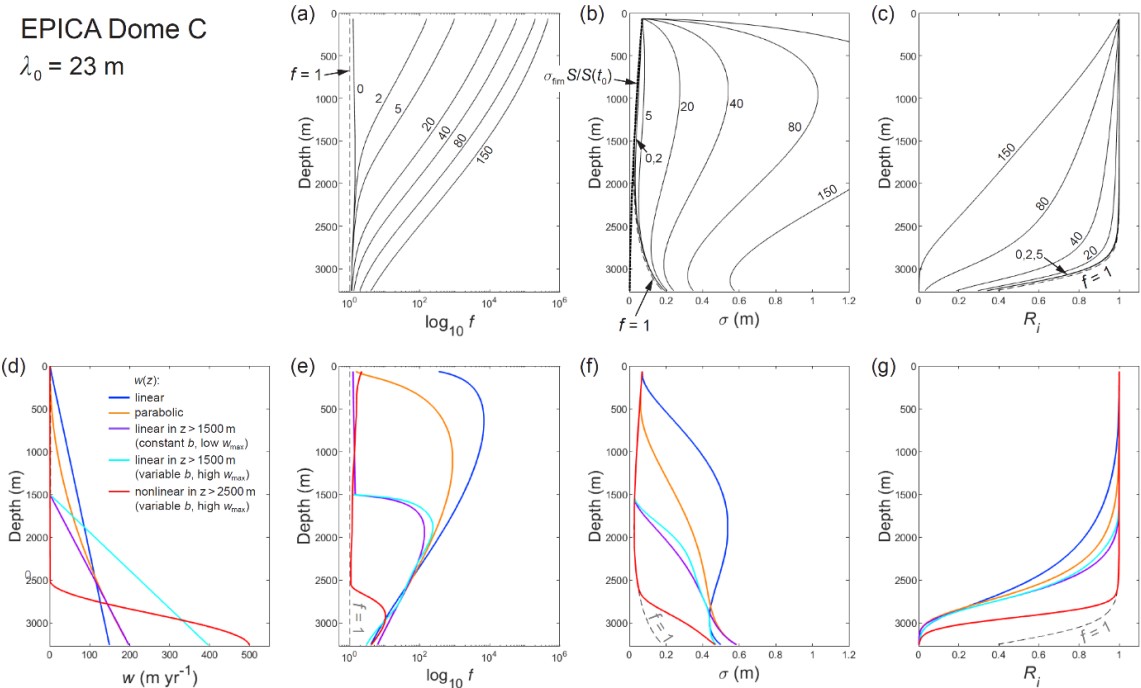

Figure 10. Computed depth profiles for the EPICA Dome C core site of (a, e) enhancement factor $f$, (b, f) diffusion length $\sigma$, and (c, g) signal amplitude ratio $R_i$. All model runs investigate a millennial-scale signal with $\lambda_0 = 23$ m yr$^{-1}$ (1000 × surface accumulation rate). In (a) to (c), each run assumes $b = 2$ mm and constant vein-water flow velocity $w$ (value beside each curve, m yr$^{-1}$). Curves labelled $f = 1$ show results based on Ramseier's diffusivity. Panels (e) to (g) report five model runs able to yield $\sigma$ of 0.4–0.6 m near the base, assuming the variable $w$-profiles in panel (d). Three runs assume $b = 2$ mm, and two runs assume the variable grain-radius profile in Fig. 8h.


we further consider that Pol et al. (2011) estimated the diffusion length $\sigma \sim 8$ cm for ice at MIS 11 (~395 to 426.7 ka, ~2699 to 2799 m) in the EPICA core, then the deep nonlinear $w$-profile (red) best mimics the observations. In any case, excess diffusion unassisted by high vein-water flow at depth cannot explain them. The limited excess diffusion at $w = 0$, which hardly

alters the signal evolution predicted by Ramseier's diffusivity (Fig. 10), is far from sufficient.

In summary, for both the GRIP and EPICA cores, vein-flow modulation with suitable choices of $w$ can reproduce the levels of excess diffusion inferred from their isotope records. Nye's model (which is approached at high $w$ in Figs. 9 and 10, as $w \rightarrow \infty$ regains his model; Sect. 3) and Rempel and Wettlaufer's model ($w = 0$) overpredict and underpredict those levels, respectively, in simulations using grain sizes similar to those measured. The reason Rempel and Wettlaufer's model revises

down Nye's enhancement factor $f$ was explained in Sects. 2 and 3. Accounting for vein tortuosity (by lowering $D_v$) weakens the short-circuiting and reduces $f$ further and does not alter these findings.


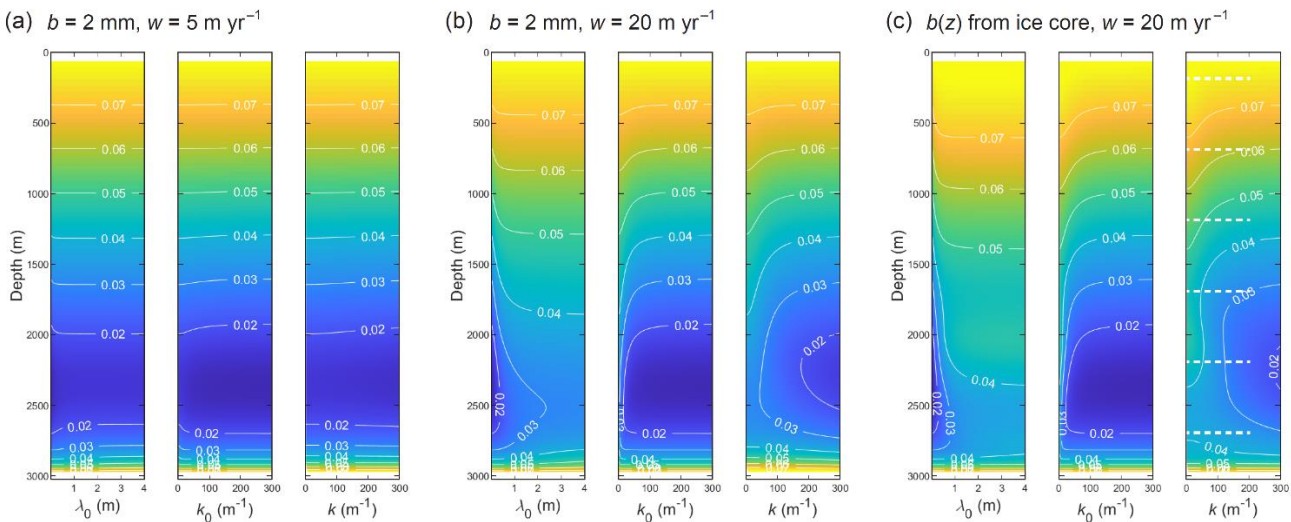

Figure 11. Wavelength dependence of diffusion length $\sigma$ (colour, contours in m yr$^{-1}$) at the GRIP site for three parameter settings: (a) $w = 5$ m yr$^{-1}$, $b = 2$ mm, (b) $w = 20$ m yr$^{-1}$, $b = 2$ mm, and (c) $w = 20$ m yr$^{-1}$ with the grain-radius profile in Fig. 8d. In each panel, the maps plot $\sigma$ versus depth against the initial wavelength of the signal $\lambda_0$, its initial wavenumber $k_0$, and its wavenumber $k$ at depth. White dashed lines on the map at far right locate the transects of Fig. 12b.

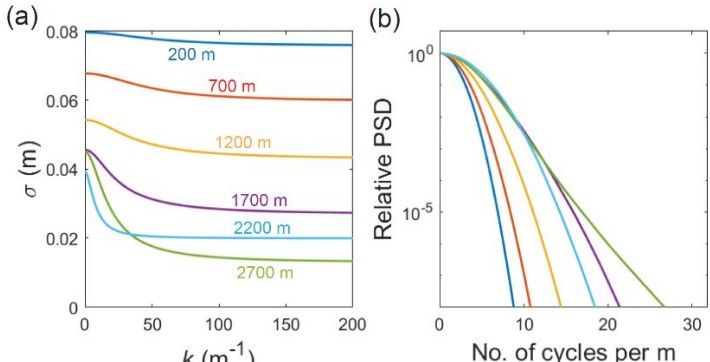

Figure 12. (a) Diffusion length $\sigma$ as a function of signal wavenumber $k$, computed at six depths in the GRIP ice column (numbers in colour) in the model runs in Fig. 11c, assuming $w = 20$ m yr$^{-1}$ and the variable grain-radius profile. (b) Synthetic power spectral density (PSD) decays based on the curves in (a), colour-coded as there. Each signal with wavenumber $k$ has $k/2\pi$ cycles per metre, so the horizontal axes in (a) and (b) are equivalent.



And what of the implications for firn temperature reconstructions? Excess diffusion can bias them in two ways, which

we discuss using the GRIP runs as an example. First, it introduces uncertainty and bias to $\sigma_{obs}$ found by PSD fitting. Figure 11

maps the computed diffusion length versus depth for signals of *different* initial wavelength $\lambda_0$ (think of the left-most plot of

each panel as showing multiple $\sigma$-profiles in a stack); in the same way, it maps $\sigma$ against their initial wavenumber $k_0 = 1/\lambda_0$

and their wavenumber at depth after accounting for thinning, $k = k_0 S(t_0)/S$. Whether the simulations use a constant or variable

grain radius, $\sigma$ develops a pronounced wavelength dependence at $w = 20$ m yr$^{-1}$ (the velocity yielding the level of excess

diffusion in GRIP Holocene ice) at most depths, except near the surface and base (Fig. 11b, c). As anticipated, $\sigma$ increases

with $\lambda_0$ and decreases with $k$ monotonically. Its variations (> 15 % below 700 m; Fig. 12a) mean that the PSD decays [$\propto$ exp(–

$k^2 \sigma^2(k)$)] are not parabolic in $k$ as they might seem (Fig. 12b). Consequently, $\sigma_{obs}$ found by fitting exp($-k^2 \sigma_{obs}^2$) to the decays

will be misestimated, and, as explained earlier, biased too high for existing firn-temperature reconstructions (by an amount

dependent on the fitting method).

This issue will arise wherever excess diffusion occurs, and in deeper ice below any section with excess diffusion. It will

affect reconstructions using the difference between the diffusion lengths of oxygen and deuterium (Simonsen et al., 2011;

Holme et al., 2018) as well as those using the diffusion length of a single isotope. It can be diagnosed by a statistically-

significant negative trend in log(PSD)/$k^2$ over $k$, although real PSD data contain noise and artefacts related to the ice-core

isotopic measurements (e.g., Kahle et al., 2018) that may complicate such a test.

Second, excess diffusion affects the calculation of $\sigma_{firn}$ in Eq. (23) – via the magnitude of $\sigma_{ice}$. Temperature reconstructions

based on diffusion length (Gkinis et al., 2014; Holme et al., 2018; Gkinis et al., 2021; Kahle et al., 2021) typically use

Ramseier's diffusivity to find $\sigma_{ice}$ (i.e., Eq. (22) with $f = 1$), which is justified for ice cores where excess diffusion does not

operate. But where it does operate, $\sigma_{ice}$ is underestimated, and $\sigma_{firn}$ and the reconstructed firn temperature are overestimated;

accounting for excess diffusion would yield a lower temperature. Put another way, a high $\sigma_{obs}$ at a given depth may result from

a large (thinned) $\sigma_{firn}$ contribution from high surface/firn temperature in the past, or from excess diffusion that raised $\sigma_{ice}$, or

from both. Robust inversion for $\sigma_{firn}$ must therefore ascertain the amount of excess diffusion.

The impact on $\sigma_{firn}$ in single-isotope reconstructions can be gauged using the GRIP runs. Figs. 9c and 9g show that the

underestimation of $\sigma_{ice}$ by Ramseier's diffusivity is minimal at $w = 0$, but reaches $\sim 0.01$ to $0.02$ m (to $\approx 2000$ m depth) if $w =$

20 m yr$^{-1}$. To gauge how much $\sigma_{firn}$ is perturbed, we calculate $\sigma_{firn}$ in Eq. (23) by using $\sigma$-profiles simulated in forward runs

515

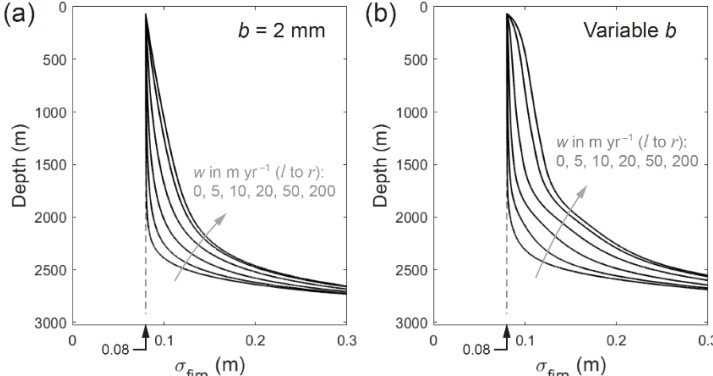

Figure 13. Firn diffusion length $\sigma_{\text{firn}}$ recovered using Eq. (23), with $\sigma_{\text{obs}}$ taken from the diffusion-length profiles simulated in forward runs at the GRIP site (for scenarios with excess diffusion at different water-flow velocities $w$), and with $\sigma_{\text{ice}}$ calculated by assuming Ramseier's diffusivity for the bulk ice. The forward runs are based on $b = 2$ mm in (a), and on the measured (variable) grain-size profile in (b).

at different $w$ as the input for $\sigma_{\text{obs}}$, but using the profile of $\sigma_{\text{ice}}$ at $f = 1$ as the other input. These tests thus imitate Ramseier-based reconstructions applied to isotope records influenced by excess diffusion (the tests are not comprehensive, as our runs used a fixed (0.08 m) rather than time-varying $\sigma_{\text{firn}}$). The results show that $\sigma_{\text{firn}}$ is overestimated by an amount increasing with depth and $w$ (Fig. 13). When $w = 20$ m yr$^{-1}$, $\sigma_{\text{firn}}$ is too high by $\sim 0.02$ m between 1000 m and 2000 m depths (ca. 5.5–17 ka) and the overestimation increases from $\sim 0.01$ to 0.03 m through this depth range. With the typical sensitivity of the firn-temperature inversion for Greenland (fig. S2 of Gkinis et al., 2014; fig. 3 of Gkinis et al., 2021), a deviation in $\sigma_{\text{firn}}$ of $\pm 0.02$ m around 0.08 m translates to a temperature change of $\approx \pm 5$ °C. Hence the firn temperature in this scenario will be overestimated by Ramseier-based reconstructions by several degrees, increasing (with age) through the Holocene and the period since the Last Glacial Maximum.

In this connection, Gkinis et al. (2014) reconstructed a temperature history from $\delta^{18}$O in the NGRIP ice core (retrieved 325 km NNW of GRIP), which they regarded as $\approx 3$ to 5 °C too warm from $\sim 8$ to 12 ka, when compared to other reconstructions (see their Fig. 6). They down-adjusted their temperature results by modifying the thinning function $S$ to reflect lower past accumulation rates. Although the glaciological conditions at NGRIP and GRIP differ, our results show that adjustments of this size are possible with vein-flow modulated excess diffusion. This is not to say that excess diffusion with $w \sim 20$ m yr$^{-1}$ occurred at NGRIP – we are not aware of reports of excess diffusion for that core. However, we recommend that


diffusion-based temperature reconstructions assess whether their results could be biased by excess diffusion, besides considering the choices and uncertainties related to firn modelling (Gkinis et al., 2021) and diffusion-length estimation (Kahle

et al., 2018). Our test case here with $w = 20$ m yr$^{-1}$ at GRIP is exemplative. Any bias will depend on the magnitude, distribution and temporal variations of $w$.

Dual-isotope reconstructions should be less affected by this problem. These reconstructions exploit the different diffusion lengths of $\delta^{18}$O (or $\delta^{17}$O) and $\delta$D in the firn, as caused by the different fractionation coefficients ($\alpha$) for $^{18}$O–$^{16}$O (or $^{17}$O–$^{16}$O) and D–H; specifically, they use the square differential $\Delta\sigma^2 = \sigma^2(\text{oxygen}) - \sigma^2(\text{deuterium})$ as the proxy for firn/surface

temperature (Simonsen et al., 2011; Holme et al., 2018). If the profiles of $\sigma_{\text{ice}}$ were identical for oxygen and deuterium, as implied by our model of excess diffusion with $\alpha \equiv 1$ (and by Ramseier-based models), then neither the size of $\sigma_{\text{ice}}$ nor bias on $\sigma_{\text{ice}}$ matter, because $\sigma_{\text{ice}}^2$ cancels out in the differencing. But the cancellation is imperfect because oxygen and deuterium differ slightly in their fractionation coefficients. To study the effect, we repeated the GRIP runs by using their $\alpha$-values in Eq. (15), to compute the corresponding $\sigma_{\text{ice}}$-profiles and the ice part of the differential, $\Delta\sigma_{\text{ice}}^2 = \sigma_{\text{ice}}^2(\text{oxygen}) - \sigma_{\text{ice}}^2(\text{deuterium})$. When

$w \sim 10$ to $50$ m yr$^{-1}$, $\Delta\sigma_{\text{ice}}^2$ reaches $\sim 10^{-5}$ m$^2$ mid-column (Fig. S3). Since the observed variations in $\Delta\sigma^2$ for Central Greenland fall in the range $\sim 10^{-4}$–$10^{-3}$ m$^2$ (e.g., figs. 2 and 3 of Simonsen et al., 2011), the ice contribution to the differential can bias dual-isotope reconstructions slightly where excess diffusion operates.

We have not repeated the foregoing analyses for the EPICA ice column, because information about its pattern of excess diffusion, limited to the $\sigma$ estimates for MIS 19 and MIS 11 (Pol et al., 2010; Pol et al., 2011), is less complete than at GRIP.

## 5 Conclusions and outlook

In this paper, we described a mechanism whereby vein-water flow amplifies the short-circuiting conceived by Nye (1998), enhancing the rate of isotopic diffusion in polycrystalline ice above the rate predicted by Rempel and Wettlaufer's (2003) model. Our simulations demonstrate its profound impact on signal smoothing in ice where the vein-water flow velocity $w$ reaches $\sim 10^1$–$10^2$ m yr$^{-1}$. We explained why vein-flow modulated excess diffusion biases the spectral estimation of diffusion

lengths from isotope records, as well as diffusion-based palaeothermometry at ice-core sites. Our findings contribute insights essential for developing more robust interpretation of ice-core isotope records.

Potential modulation of excess diffusion by vein-water flow means that ice-core isotopic signals may have been altered in more complex ways than previously thought. Where modulation occurs, neither Ramseier's (1967) formula nor Rempel and Wettlaufer's (2003) model describe the bulk-ice isotopic diffusivity, and we caution their use in ice-core analysis.



Our GRIP and EPICA simulations (Sect. 4) represent the first detailed exploration of the impact of excess diffusion on diffusion-length profiles and probe the conditions behind the levels of excess diffusion inferred for those cores. Their expository nature should be emphasised: we have not fitted the observations precisely, and the vein-flow velocities in our model runs are only trial values in a sensitivity analysis. One reason is that the present model idealises some geometrical aspects of the system. Based on Nye's and Rempel and Wettlaufer's set-up (Fig. 1), it approximates the vein cross-section –

which consists of three convex faces (Nye, 1989; Mader, 1992a) – as circular, and it assumes that veins are static features, even though grain-boundary migration in ice implies a continual slow motion of veins relative to crystal matrices (Ng, 2021), which will perturb the isotope concentration fields within ice grains. Another reason is that the model ignores isotope diffusion along grain boundaries (Johnsen et al., 2000), which might increase the amount of excess diffusion in fine-grained ice (Jones et al., 2017). Moreover, two factors in the simulations are difficult to constrain, namely (i) the suppression of short-circuiting

due to blockage of veins by dust particles and bubbles and (ii) spatial variation in the vein radius. Regarding the latter, the mean vein size in ice at thermodynamic equilibrium should be governed locally by the average grain size, temperature, and total amount of dissolved ionic impurities in the veins (Nye, 1991; Mader, 1992b; Dani et al., 2012). Factors (i) and (ii) suggest possible influences by changing vein-impurity signals (Ng, 2021) and changing distributions of bubbles and solid particles on excess diffusion and the smoothing of isotopic signals. Extending the model for these controls and the detailed geometry of

the vein network as grain size and texture evolve are worthwhile avenues for further research.

A striking realisation from our study is how little is known about the vein-scale hydrology of ice sheets. The vein-flow velocity $w$ is needed for predicting excess diffusion with the model, or validating model runs matching diffusion lengths measured from ice cores. However, reliable prediction of the size and pattern of $w$ at ice-core sites is out of reach; our only handle on $w$ is Nye and Frank's (1973) theory for glaciostatically-driven porewater flow. This theory calculates the rate of

(Poiseuille) flow through veins under the hydropotential gradient $(\rho_w - \rho_i)g$, where $\rho_w$ is water density, $\rho_i$ is ice density and $g$ is gravitational acceleration. It yields a large range of plausible $w$ because the ice porosity – a key input that determines the vein size for a given mean grain size – is uncertain for polar ice. When modelling the profile of $w$ above subglacial lakes, Rempel (2005) combined Nye and Frank's (1973) theory with an equation of vein-equilibrium thermodynamics to constrain the porosity through the influence of dissolved impurities alluded to above. Although this approach can predict $w$ specifically,

it requires knowledge (or assumption) about the amount of ionic impurities partitioned to the vein network, which is not resolved by most ice-core analytical measurements. Besides, neither his model nor Nye and Frank's model has been observationally tested. A separate theory by Nye (1976) treats vein-water flow in detail, but analyses its stability only, not flow rates. Thus currently, knowledge about $w$ in polar ice is limited to a few unverified theories, and we cannot assess whether the range of $w$ found to modulate excess diffusion sensitively are common at ice-core sites.

Direct measurements of vein-water flow in polar ice are critical for progress, for informing studies of excess diffusion and hydrological modelling. Vein size measurements at low temperature are also desirable, because the observations of Mader (1992b) (vein widths ~10–100 $\mu$m) were made within 1 °C below the melting point. It may be possible to measure $w$ by



innovating on NMR and Doppler-based techniques. Laboratory studies should couple vein size and flow measurements with experiments where water percolates through ice containing isotopic signals. In such instances, one can use LA-ICP-MS (laser

ablation inductively-coupled plasma mass spectrometry; Bohleber et al., 2021) to try to discern excess diffusion, by studying the relationship between isotopic signals in the crystals and those in the veins, and looking for the flow-induced "shear layer" in isotopic concentration near triple junctions predicted by us (Sect. 3). Hitherto, no experiments have been made to demonstrate Nye's short-circuiting effect, even for the case without water flow.

Because the amplification of excess diffusion is due to isotopic gradients caused by vein-flow advection (Sect. 3), our

model implies that any non-zero pattern of $w$ will accelerate the smoothing of isotopic signals. The water flow need not be vertical or unidirectional, nor occur on long length scales (as experimented herein). We think that recrystallisation in deforming ice will cause nonuniform vein-water flow at the grain scale, although no theories yet address this process and we do not know the flow velocities involved. Notably, polygonisation and strain-induced migration recrystallisation that reconfigure grain boundaries must create new vein segments while eliminating others, thus inducing local water flow superposed on any long-

range transport (e.g., the downward percolation envisaged by Nye and Frank (1973)). If this hypothesis is correct, then the coupling between recrystallisation processes and isotopic diffusion is more complicated than the irregular shape and motion of existing veins, and the rate and mode of ice deformation will affect the level of excess diffusion.

Future studies should analyse the pattern of excess diffusion in multiple ice cores systematically, to help unravel its diverse controls; this is not least because the origin of excess diffusion at the GRIP and EPICA sites remains elusive. High-

resolution isotopic measurements on ice cores that are becoming more common (e.g., Steig et al., 2021) will aid this effort. Our modelling shows that where excess diffusion occurs, one should be able to quantify and remove the biases on spectrally-derived diffusion lengths and firn-temperature reconstructions, by calculating $\sigma_{ice}$ and its wavelength dependence with Eqn. (22). Although this is not possible to do until $w$ can be predicted for a given core site, diffusion-based studies should at least scrutinise their isotope records for signs of excess diffusion, by studying the decay rate of signals at specific (unthinned)

wavelengths (Johnsen et al., 1997; Johnsen et al., 2000) and testing for significant trends on $\log(PSD)/k^2$ data (Sect. 4).

**Acknowledgements**

I thank Andrew Sole and Adam Hepburn for comments on the manuscript and Greg Kimmel for providing the diffusivity data from the study by Xu et al. (2016).

**Code and data availability**

The MATLAB code for evaluating the model and computed data are archived at https://doi.org/10.15131/shef.data.xxxxxxxx. Please use https://figshare.com/s/dc30685fedcdeedc66a0 during the review stage.



## Supplement

Figures S1–S3 and Movies S1–S3 are available at https://doi.org/10.15131/shef.data.yyyyyyyy. The supplement related to this
article is available online at https://doi.org/10.5194/tc-zz-zzzz-zzzz-supplement.

Please use https://figshare.com/s/79e62bdae0f84a2efa11 during the review stage.

## Author contribution

F. S. L. Ng designed the study, performed all analyses, and wrote the paper.

## Competing interests

The author declares no conflict of interest.

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



## Appendix A: diffusivities $D_v$ and $D_s$

Following Rempel and Wettlaufer (2003), we use a relation for $D_v(T)$ based on experimental measurements on supercooled
water, even though strictly speaking the vein water is liquid – in thermodynamic equilibrium with ice – due to the presence of
dissolved impurities and interfacial curvature (Mulvaney et al., 1988; Nye, 1991; Ng, 2021), not because of supercooling.

Rempel and Wettlaufer (2003) fitted a quadratic to the self-diffusion coefficients measured by Gillen et al. (1972) with
an NMR method for supercooled water down to $\approx$ –31 °C. At lower $T$, their quadratic is not meant to apply, especially as it
predicts $D_v$ to increase again (Fig. 2a, red dashed curve). One cannot extrapolate the trend of Gillen et al.'s (1972) data to those
temperatures, as there has been much debate about a possible liquid–liquid phase transition in the so-called 'no man's land' of
deeply supercooled water near 228 K ($\approx$ –45 °C), which is thought to be responsible for various observed anomalies in the
properties of water. Reviews of this topic have been given by Amann-Winkel et al. (2016), Handle et al. (2017) and Hestand
and Skinner (2018).

Recently, Xu et al. (2016) measured the rate of growth of ice crystals into supercooled water by using a pulsed-laser
heating technique and used the results with the Wilson–Frenkel model (Wilson, 1900; Frenkel, 1946) to derive new estimates
of $D_v$, down to 125 K. From their growth and diffusivity data, they inferred no thermodynamic transitions or singularities in
'no man's land'. Here we fit their $D_v$ values between –12.8 °C and –60.8 °C (blue circles, Fig. 2a), which exhibit a slope break
on the Arrhenius plot at $\approx$ –38 °C, by using the composite exponential

$$D_v = \cfrac{1}{\cfrac{1}{1.085\times10^{-6}\,\exp\!\left(\cfrac{-1870}{T}\right)} + \cfrac{1}{2.942\times10^{7}\,\exp\!\left(\cfrac{-9474}{T}\right)}} \quad \text{m}^2\,\text{s}^{-1} \qquad\qquad \text{(A1)}$$

(blue curve, Fig. 2a). Although their experiment was conducted in an ultra-high vacuum, we use (A1) without pressure
correction in our model, because combined atmospheric and glaciostatic pressures should increase $D_v$ by a few percent only
($D_v$ increases by 10–40 % as pressure rises from 0.1 to 100 MPa (Prielmeier et al., 1988)) and because the data points of Xu
et al. actually lie above those of Gillen et al. (Fig. 2a).

For the self-diffusivity of ice, $D_s$, we use Ramseier's (1967) formula

$$D_s = 9.1\times10^{-4}\,\exp\!\left(-\frac{7.2\times10^{3}}{T}\right) \quad \text{m}^2\,\text{s}^{-1}, \qquad\qquad \text{(A2)}$$

which is based on experiments on ice monocrystals down to –35.9 °C. We are not aware of measurements of $D_s$ at lower
temperatures, where diffusion is too slow to be easily determinable on experimental timescales. However, there is no reason
to expect that the activation energy of volume self-diffusion ($\approx$ 60 kJ mol$^{-1}$ according to (A2)) would change drastically at low
$T$. Examples of the use of (A2) for $T$ as low as $\approx$ –50 °C appear in earlier studies (e.g., Pol et al., 2010; Grisart et al., 2022).





780

## Appendix B: scaled model

By non-dimensionalising the independent variables with $r^* = r/b$, $z^* = z/b$ and $t^* = t/(b^2/D_s)$, Eqs. (3) to (5) become

$$\frac{\partial \delta}{\partial t^*} = \frac{1}{r^*}\frac{\partial}{\partial r^*}\left(r^*\frac{\partial \delta}{\partial r^*}\right) + \frac{\partial^2 \delta}{\partial z^{*2}} \,,$$ (B1)

with the boundary conditions

785
$$\frac{\partial \delta}{\partial r^*} = 0 \quad \text{at} \quad r^* = 1 \,,$$ (B2)

$$\frac{\partial^2 \delta}{\partial r^{*2}} - \frac{1}{\varepsilon}\frac{\partial \delta}{\partial r^*} - \beta\frac{\partial^2 \delta}{\partial z^{*2}} + \gamma\frac{\partial \delta}{\partial z^*} = 0 \quad \text{at} \quad r^* = \varepsilon \,.$$ (B3)

The parameters

$$\varepsilon = \frac{a}{b} \quad (\ll 1) \qquad \text{and} \qquad \gamma = \frac{wb}{D_s}$$ (B4)

signify the dimensionless vein radius and the dimensionless vein-water flow velocity, respectively. If we rescale the decay-

790 rate parameter, wavelength and wavenumbers according to $\zeta^* = b^2\zeta$, $\lambda^* = \lambda/b$ and $[k_z^*, k_r^*] = b[k_z, k_r]$ (thus ensuring $\lambda^* = 2\pi/k_z^*$), then the solution in Eq. (7) becomes

$$\delta(r^*, z^*, t^*) = \delta_0 + \delta_1 F(r^*)\exp(-\zeta^* t^* + ik_z^* z^*) \,,$$ (B5)

and the enhancement factor $f = 1 + (k_r^*/k_z^*)^2$ retain the form in Eq. (9).

This scaled model implies $f = f(\varepsilon, \lambda^*, \gamma, \beta)$ or $f(\lambda^*, \gamma, \beta)$ at fixed $\varepsilon$. The signal migration velocity $v$ ($= \zeta_I D_s/k_z$

795 dimensionally) is given by $(D_s/b)\mathrm{Im}(\zeta^*)/k_z^*$ or $(D_s/b)g$, where $g$ is another function of the same parameters. Indeed, one can

solve the scaled model by the method of Sect. 2 (replacing $a$ by $\varepsilon$, $b$ and $D_s$ by 1, and $w$ by $\gamma$) and, after computing $f$ and $g$,

infer the dimensional controls on $f$ and $v$ (considered in Sect. 3 and Fig. 7) through the parameter and scaling definitions.