# Peer review of "Isotopic diffusion in ice enhanced by vein-water flow"

_The Cryosphere, 2023_

## Referee Comment (RC1)

Ng, "Isotopic diffusion in ice enhanced by vein-water flow"

The main achievement of this work is the introduction of more physics to the study of "smoothing" of isotopic signals due to diffusion in the study of ice. Previous work by others has introduced the concept of enhanced diffusion (compared to that within the ice crystals) due to liquid in veins in the ice. This work adds an additional plausible factor, which is the transport of isotopes due to flow of liquid within the veins.

The model of a single uniform pore is, of course, greatly oversimplified, but is enough to capture the basic physics being studied. The author nicely shows that some previous work is recovered in the limits of zero flow and high flow velocity. Various cases of the model are run to demonstrate the effect of flow on quantities such as effective diffusion lengths, including for two real ice-core sites. The case is made that inclusion of flow will be necessary for proper isotopic modeling in some cases where excess diffusion occurs. The author rightly notes the need for better knowledge of veins and liquid flow within ice.

My main expertise (and the aspect I was asked to review) is in thermophysical data for water and ice, not in ice core analysis. So my review is focused on some of the assumptions made and used in the model.

But I want to begin with an important caveat. In several places I will point out areas where input parameters to the model (such as diffusivities) are oversimplified or have more uncertainty than one would glean from reading the manuscript. I am NOT suggesting that the author do more calculations to account for these extra factors in the current paper. This is an expository study, not intended to quantitatively model ice cores. It seems very likely that the basic findings of the work would be the same if, for example, a diffusivity was different by a factor of 2—it might just mean a small change in flow rate or pore size or addition of a tortuosity factor to achieve a similar numerical result. Instead, my suggestion is that some of these factors could be mentioned around line 580, where the author already mentions other factors that would need to be better described to rely on this approach for quantitative modeling. Probably some of the things I will mention are completely insignificant within the model (presumably the author can tell how sensitive the model is to various factors), so it would be fine to omit mention of such items completely.

With that said, here are some potential opportunities for improvement in the data used.

The diffusivity in crystalline ice appears to be enhanced by pressure, as found by Noguchi et al., https://doi.org/10.1016/j.pepi.2016.05.010. From a graph in the paper, it looks like about an order of magnitude effect at their pressure of 100 MPa (which is beyond that of the ice cores studied here), so if that work is correct this may be significant (say, at the level of a factor of 2) for deep ice cores. As noted in Appendix A, the pressure dependence of the diffusivity of liquid water appears to be small at pressures relevant to any ice that would be encountered on Earth (see also Harris & Newitt, https://doi.org/10.1021/je9602935).

The use of the diffusivity for pure liquid water is an approximation if the water in the veins is in the liquid state due to dissolved ionic impurities. I do not know what the concentration of the ions is believed to be, but if it is significant (which it must be to have stable liquid at the temperatures studied here) that would significantly affect the water's diffusivity. The effect of

dissolved NaCl on the diffusivity of water at low temperatures was reported by Garbacz and Price (https://doi.org/10.1021/jp501472s).

The dissolved ions in the water will also affect the solid-liquid fractionation (maybe this can be ignored since the author makes an $\alpha$=1 simplification in most of the work). Salt effects on isotopic fractionation have been studied (for vapor-liquid fractionation, which I would expect to be similar to the effects on S/L fractionation because the salt only affects the liquid-phase thermodynamics), see for example the work of Horita et al. and references therein (https://doi.org/10.1016/0016-7037(95)00031-T), but the effect is probably too small to matter in the current context.

There is more uncertainty than the paper suggests in the diffusivity of pure liquid water at the relevant conditions. A curve-fit of data from Gillen (1972) is used, but the more recent work of Price et al. (https://doi.org/10.1021/jp9839044) gives diffusivities up to 30-40% higher [the Price data were judged to be uncertain within about 5% in a later evaluation (see supporting information of https://doi.org/10.1073/pnas.1508996112)]. Then the range below about 240 K depends entirely on the work of Xu et al., which is an indirect method based on a model of crystal growth that is reasonable but by no means certain. The degree to which the uncertainty in liquid diffusivity is problematic and should be mentioned depends on how sensitive the model is to this parameter.

More recent recommended values for solid-liquid fractionation factors were reported by Wang and Meijer, https://doi.org/10.1080/10256016.2018.1435533. However, their results are consistent with the older source used here. The solid-liquid factors are only measured at 0 °C, and they must be at least weakly temperature dependent. But using the 0 °C values at all conditions is the reasonable approach given the absence of data at other temperatures.

There is also the fact that the isotopes will not diffuse at the same rate in liquid water, so the heavy isotopes will not diffuse quite as quickly in the vein as the model assumes. However, I think that a factor of (to a zeroth approximation) SQRT(18/19) or SQRT(18/20) would be negligible in this context unless there is some amplification mechanism I am not thinking of.

---

## Author Response (AR1)

6th June 2023

Dear *Editor* Dr. Casado:

Please find attached my revised manuscript – both its final and track-change versions.

Following the suggestions of Reviewer 1 and Reviewer 2, respectively, I have extended the *Discussion* to include two passages to

(i)      explain [on Lines 575-583] that the diffusivities $D_v$ and $D_s$ are subject to uncertainty and potentially influenced by other factors, to let the reader know that the recipes currently used for them are not necessarily accurate nor comprehensive [the long paragraph containing the passage has also been edited for flow and concision];

(ii)     outline the need to establish whether the theorised phenomenon really operates [Lines 623-629], including Reviewer 2's idea about studying existing ice-core records for any displacement of isotopic signals due to the signal migration predicted by the theory.

These revisions follow what I agreed to do in my author responses during the interactive open discussion.

With regard to (i), Reviewer 1 kindly provided various details and examples (with references) on what can affect the diffusivities and fractionation factors. However, I have decided not to add such details to *Appendix A*, because I find I can't add one without adding all of them, yet the whole set of details remains highly fragmentary, given the state of current understanding.

Here and there, minor corrections have been made to the text to improve clarity and emphasis or remove redundancy/clutter. Where previously I used the wrong unit m yr$^{-1}$ for the initial signal wavelength $\lambda_0$ (e.g. on Lines 406 and 446), the unit has been corrected to metre (a length). This correction has been made also to the units inside the key in Fig. 9e.

Fig. 10 has been replaced with a new version without the spurious grey "0" (next to the vertical axis) in panel 10d.

The DOIs of the data/code repository and supplementary material are now known. I indicate them on Lines 645, 648 and 730.

Following an open-access requirement by my university, in the *Acknowledgements* I have added a statement on Lines 641-643 to declare the CC-BY licence for any Author Accepted Manuscripts, using the precise wording given by my university.

On the next page, I outline some changes that I wish to make to improve the "*Short Summary*" on the TC webpage of the manuscript. Please kindly ask the editorial office to update the summary.

Thank you for your attention and for handling my manuscript.

Kind regards,
Felix Ng

**Isotopic diffusion in ice enhanced by vein-water flow**

**Short Summary**

Original version, with edits in red:

The stable isotopes of oxygen and hydrogen in ice cores  are routinely analysed for the climate signals which they carry. It has long been known that the system of water veins in ice facilitates isotopic diffusion. Here, mathematical modelling shows that water flow in the veins strongly accelerates the diffusion and  the decay of climate signals. The process hampers methods that use the variations in signal decay with depth to reconstruct past climatic temperature.

Corrected version:

The stable isotopes of oxygen and hydrogen in ice cores are routinely analysed for the climate signals which they carry. It has long been known that the system of water veins in ice facilitates isotopic diffusion. Here, mathematical modelling shows that water flow in the veins strongly accelerates the diffusion and the decay of climate signals. The process hampers methods that use the variations in signal decay with depth to reconstruct past climatic temperature.

468 characters

---

## Author Response (AR2)

21st June 2023

Dear *Editor* Dr. Casado:

Many thanks for the good news that you recommend acceptance of my manuscript.

Please find attached my finalised manuscript, where I have made the requested correction to replace the word "recipes" by "formulae" on Line 576 (or Line 582-3 on the track-change file of the last round).

I have also observed and re-checked the colour schemes of all figures (in both main and supplementary materials) that they abide with TC's policy for readers with colour vision deficiencies.

The only challenging figure in this regard was Figure 7, where two colour schemes are used, one for panels a, b, d, e, and the other for panels c and f. I have checked that both colour schemes are differentiated when checked again the Coblis simulator for readers with different vision conditions. An issue arises only if the reader has no colour vision at all – sees everything as *monochromatic*. However, that is why all panels are plotted in 3D (so the monochromatic reader can gauge heights above the plane of zero datum, as much as a reader with normal vision can do), and it is why the panels aren't plotted in 2D as planimetric maps (i.e. looking down vertically onto the surfaces).

Should there be any further technical corrections, I would be happy to tackle them.

Thank you again for your consideration.

Kind regards,
Felix Ng